# CYFIP1 governs the development of cortical axons by modulating calcium availability

Carlotta Ricci[1,2], Maëllie Julie Midroit[1], Federico Caicci[3], Tilmann Achsel[1], Nuria Domínguez-Iturza [1,2,4,5] ✉ & Claudia Bagni [1,2,5] ✉

The human CYFIP1 gene is linked to Autism Spectrum Disorder (ASD) and Schizophrenia (SCZ), both associated with brain connectivity defects and corpus callosum abnormalities. Previous studies demonstrated that *Cyfip1*-heterozygous mice exhibit diminished bilateral functional connectivity and callosal defects—resembling observations in ASD and SCZ patients. Here, we demonstrate that CYFIP1 is crucial for cortical axonal development and identify insufficient calcium uptake as the pivotal mechanism. In vivo*, Cyfip1* heterozygosity delays callosal axon growth and arborization. Additionally, *Cyfip1*-deficient cortical neurons and axons have reduced intracellular calcium, along with impaired mitochondria morphology, activity, and motility. Mechanistically, CYFIP1 binds and stabilises the mRNA of specific voltage-gated calcium channel subunits, explaining the decreased calcium concentration in $Cyfip1^{+/-}$ cells. Notably, elevating intracellular calcium rescues delayed axonal growth and mitochondrial defects in *Cyfip1*-deficient neurons. These findings highlight that, by regulating mRNA metabolism, CYFIP1 ensures proper callosal development, offering insights into brain connectivity disruptions underlaying neurodevelopmental disorders.

Neurodevelopmental disorders (NDDs) are a group of conditions caused by disturbed brain development. Many NDDs, including Autism Spectrum Disorder (ASD) and Schizophrenia (SCZ), are characterized by abnormal development of brain connectivity, including defects in grey and white matter structure, as well as impaired functional connectivity[1–6]. These observations lead to term the diseases as developmental disconnection syndromes[7,8]. Copy number variations (CNVs) in the chromosomal region 15q11.2 have been associated with a higher risk of developing ASD and SCZ. This genomic region encodes four genes: *TUBGCP5*, *NIPA1*, *NIPA2* and *CYFIP1*. Genomic variations and point mutations identified *CYFIP1* as the most likely gene from the region causally associated with SCZ[9–11] and ASDs[12,13]. Furthermore, *CYFIP1* mRNA expression levels also positively correlated with the severity of ASD in individuals with Fragile X Syndrome[14].

The Cytoplasmic FMRP-interacting protein 1, CYFIP1, was initially described as specifically Rac-1 associated protein (Sra-1)[15], but later studies showed that it exerts a dual function. CYFIP1 is part of the Wave Regulatory Complex (WRC) and regulates ARP2/3 dependent actin polymerization downstream of Rac-1 activation[16–20]. In its alternative role, CYFIP1 binds FMRP and eIF4E repressing translation of target mRNAs[21–23]. A precise regulation of these two processes is essential for correct brain development and function[24]. Furthermore, the brain CYFIP1 interactome, revealed the presence of several RNA-binding proteins involved in other key aspects of mRNA metabolism, including mRNA stability[17].

CYFIP1 is ubiquitously expressed in the body and across the whole brain[25,26]. CYFIP1 expression starts at early embryonic stages and its homozygous deletion in mice is embryonic lethal at E8.5[17,26,27]. Several studies over the last years have investigated the effect of *Cyfip1*

[1]Department of Fundamental Neurosciences, University of Lausanne, Vaud, Switzerland. [2]Department of Biomedicine and Prevention, University of Rome Tor Vergata, Rome, Italy. [3]Department of Biology, University of Padova, Padua, Italy. [4]Present address: Department of Stem Cell and Regenerative Biology, Harvard University, Cambridge, USA and Stanley Center for Psychiatric Research, Broad Institute of MIT and Harvard, Cambridge, MA, USA. [5]These authors jointly supervised this work: Nuria Domínguez-Iturza, Claudia Bagni. ✉e-mail: nuriadomingueziturza@fas.harvard.edu; claudia.bagni@unil.ch

heterozygosity (*Cyfip1*[+/-]) in mice and found that reduction of CYFIP1 leads to ASD- and SCZ-like behavioural phenotypes[27–30], demonstrating that the *Cyfip1*[+/-] mouse model has face validity for the study of these diseases. Previous studies from our laboratory and others demonstrated the crucial role of CYFIP1 in brain connectivity, white matter architecture and axonal function. *Cyfip1* heterozygous mice (*Cyfip1*[+/-]) have reduced functional connectivity and altered presynaptic function[28,31]. In addition, *Cyfip1*[+/-] mice and rats present white matter architectural defects and reduced myelination of the corpus callosum[28,32]. While CYFIP1 has been shown to regulate the growth of retinal axons in zebrafish[33], and its role in post-synaptic development and transmission has been previously studied[17,34,35], very little is known about how CYFIP1 regulates axonal development and circuit formation. Here, we show that CYFIP1 plays a crucial role in axonal development by regulating intracellular calcium availability and mitochondrial function. In vivo, CYFIP1 regulates the axonal development of callosal projection neurons. Mechanistically, axons from *Cyfip1*[+/-] cortical neurons exhibit reduced intracellular calcium levels alongside increased mitochondrial area, motility, and density, that ultimately contributes to reduced mitochondrial activity. At the molecular level, CYFIP1 modulates the expression of several alpha subunits of voltage-gated calcium channels. Notably, elevating intracellular calcium concentration was sufficient to rescue both axonal growth and mitochondrial defects in *Cyfip1*[+/-] neurons. Together, our findings reveal the crucial role of CYFIP1 in cortical axonal development and suggest that its dysfunction may underlie the connectivity defects observed in vivo in the adult *Cyfip1*[+/-] mice and reported in individuals with ASD and SCZ.

## Results

### Axonal development in vivo is impaired in *Cyfip1*[+/-] mice

To investigate whether *Cyfip1* deficiency could compromise axonal growth in vivo, we performed in utero electroporation (IUE) at embryonic stage 15.5 (E15.5) with a plasmid expressing the tdTomato fluorescent protein. This approach allows the labeling of callosal projection neurons (CPN) located in upper layers of the cortical plate (layer II/III) which project their axons through the corpus callosum to the contralateral hemisphere (Fig. 1a). We analyzed their axonal projections at three different postnatal (P) stages: P5 during axonal growth into the contralateral hemisphere, P15 during axonal arborization, and P30 after axonal refinement[36–38] (Fig. 1; Fig. S1). We found that axonal growth into the contralateral cortex was reduced at P5 in *Cyfip1*[+/-] mice (Fig. 1a). Notably, this defect was rescued when we electroporated, together with tdTomato, the WT form of *Cyfip1* (*Cyfip1*[+/-] + cDNA *Cyfip1*), suggesting that the observed axonal growth defect may be cell autonomous. When we analyzed axonal invasion into the contralateral cortex at P15, we observed that the axons of both genotypes had nonetheless reached their target region (length equivalent to ~95% of the cortical thickness, Fig. 1b), indicating that *Cyfip1* deficiency leads to a delay in axonal growth. The delayed arrival of the axons might influence their targeting of the post-synaptic cells. We therefore analyzed the terminal arborization of the axons in the upper cortical layers (LII/III) at P15. The total number of branches in *Cyfip1*[+/-] mice was reduced compared to WT, due to a decrease in both primary and secondary branches (Fig. 1c). By postnatal day 30 (P30), axonal arborization was comparable between WT and *Cyfip1*[+/-] mice (Fig. S1) indicating that arborization is also slowed down.

Altogether, our findings show that *Cyfip1* deficiency results in delayed axonal growth and arborization, demonstrating that CYFIP1 is crucial for timely axonal development in the neocortex.

### *Cyfip1*[+/-] cortical axons display increased mitochondrial density and motility

Proper timing of axonal growth and branching during development are critical processes for proper brain connectivity and wiring, which were previously reported to be affected in *Cyfip1*[+/-] adult mice[28]. To further investigate the cellular and molecular mechanisms behind these developmental defects we opted for an in vitro system with fewer variables and focus specifically on the axonal growth phenotype. First, we investigated if the delayed axonal growth observed in vivo could be recapitulated in vitro. To do so, we used a microfluidic chamber system that allows the physical separation of cell bodies and dendrites from axons. We plated mouse cortical neurons on one side of the chamber, what we termed the soma compartment, and allowed neurons to grow for 3 days in vitro (DIV 3). At this time, only axons are long enough to transverse the microchannels and reach the opposite side of the chamber, the so-called axonal compartment (Fig. 2a). We measured the length of individual neuronal axons at DIV 3 and DIV 4 and calculated the growth rate during those 24 h. As shown in Fig. 2a, the growth of *Cyfip1*[+/-] axons was reduced compared with WT. This result recapitulated our in vivo observation and strengthened the hypothesis that CYFIP1 may regulate axonal growth in a cell autonomous manner.

It was previously shown that CYFIP1 is present in neuronal axons and growth cones[17,39], and that reduced CYFIP1 levels in *Drosophila* compromise mitochondrial function leading to social deficits[40]. Moreover, it is known that proper mitochondrial function and distribution in the axon is crucial for axonal development[41]. We therefore hypothesized that CYFIP1 may regulate axonal mitochondria leading to the observed delay in axonal growth. To test that, we transfected primary cortical neurons with a pTagRFP-mito plasmid that will fluorescently label mitochondria and investigated their axonal distribution. We found that mitochondrial density was increased in *Cyfip1*[+/-] axons compared to WT. This increase was evident both along the whole axon and in different axonal regions (proximal, medial, and distal) (Fig. 2b), while mitochondrial density along dendrites was comparable between the two genotypes (Fig. S2a). In addition, the total mitochondrial mass, measured as the mitochondrial to total DNA ratio, was similar for WT and *Cyfip1*[+/-] cortical neurons (Fig. 2c), suggesting that the increased mitochondrial density observed in the axon may be due to a transport defect. We next investigated mitochondrial motility along the axon by performing a kymograph analysis on transfected DIV 3 primary cortical neurons. We found that the axons of *Cyfip1*[+/-] cortical neurons have an increased percentage of motile mitochondria compared with WT (Fig. 2d). Several other parameters analyzed, including forward and backward movement length, forward and backward movement speed, and the frequency of reverse direction transitions, showed no significant differences between the two genotypes (Fig. S2b). Interestingly, mitochondrial transport at later developmental stages was comparable between WT and *Cyfip1*[+/-] (Fig. S2c).

These results demonstrate that reduced levels of CYFIP1 compromise axonal distribution and transport of mitochondria, emphasizing its crucial function within a specific developmental period which could contribute to the delay axonal growth phenotype previously observed in those cells.

### Intracellular Ca²⁺ homeostasis and mitochondrial activity are compromised in *Cyfip1* deficient neurons

Mitochondrial transport along the axon is a highly regulated process, in which motor proteins (kinesin and dynein) transport mitochondria in anterograde or retrograde direction, ensuring adequate supply and clearing[42]. To provide the required energy, mitochondria are docked to the actin cytoskeleton of specific axonal regions[43]. A fundamental molecule for the regulation of mitochondrial transport is $Ca^{2+}$. While the precise mechanism is still debated, an increase in calcium concentration [$Ca^{2+}$] within the cell or the mitochondrial matrix results in enhanced mitochondrial docking and reduced motility[44–50]. We hypothesized that CYFIP1 may regulate intracellular $Ca^{2+}$ levels, preventing mitochondria docking and leading to the increased number of motile

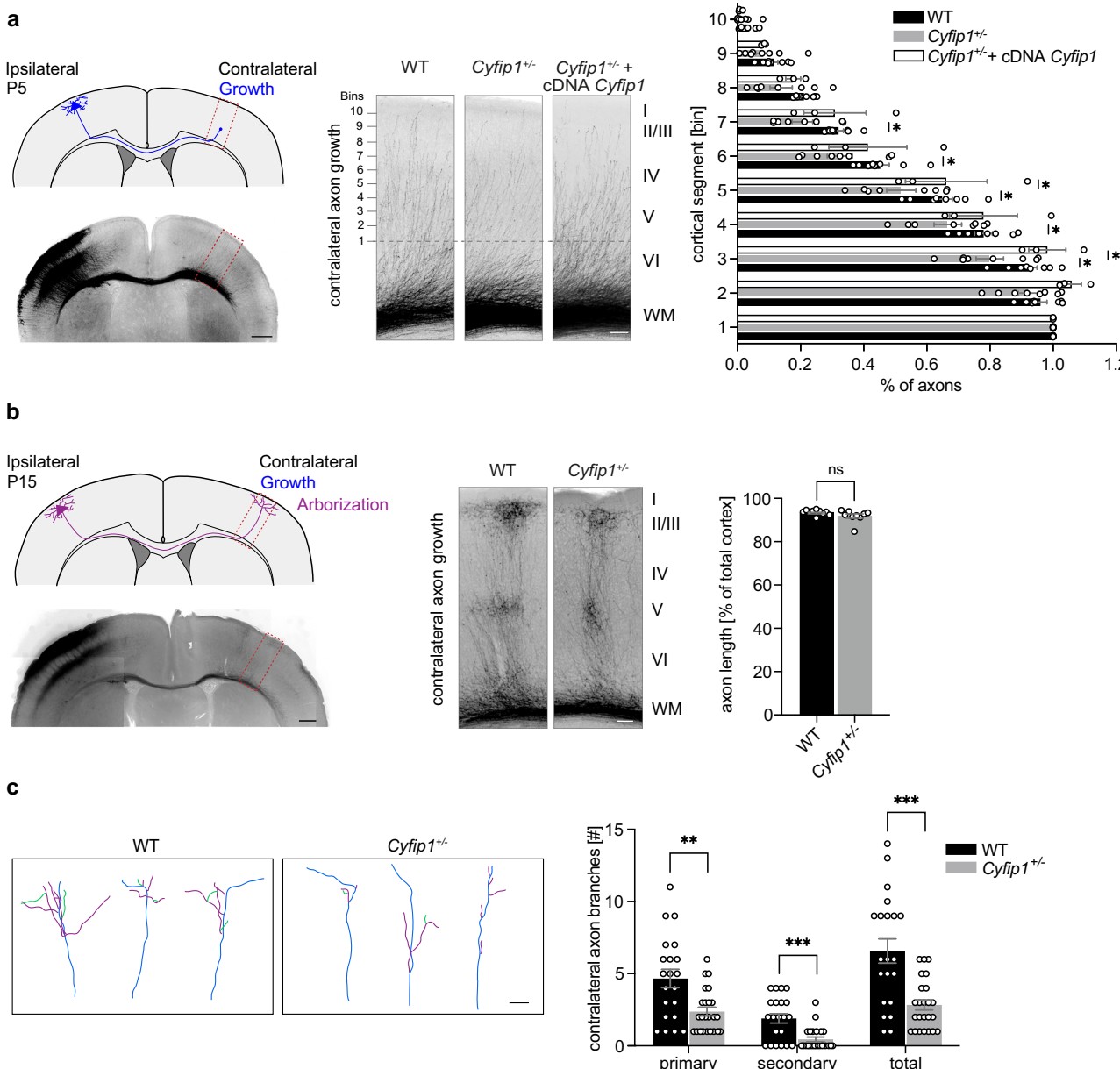

**Fig. 1 | Development of callosal projections is impaired in *Cyfip1*⁺/⁻ mice. a** Left, representative schematics and fluorescence image of a postnatal day 5 (P5) brain section after *in utero* electroporation (IUE) at embryonic stage 15.5 (E15.5) with a tdTomato-expressing plasmid (pCAG-tdTomato, shown in black). The image was assembled from multiple overlapping micrographs. Contralateral somatosensory cortex is outlined with a red box. Scale bar 500 μm. Middle, representative fluorescence images of WT and *Cyfip1*⁺/⁻ mice contralateral cortex at P5, after IUE with pCAG-tdTomato (for WT and *Cyfip1*⁺/⁻) or a mix of pCAG-tdTomato and pCAG-EGFP-hCYFIP1 WT (*Cyfip1*⁺/⁻ + cDNA *Cyfip1*). Cortical layers are outlined on the right (Layer I, II/III, IV, V, VI and white matter (WM)) and analyzed segments (bins) on the left. Scale bar 100 μm. Right, the invasion index of the axons into the contralateral cortex was quantified as the ratio of the number of axons growing into each bin, to the number of axons emerging from the corpus callosum (calculated at 300 μm from the white matter) (WT *n* = 8 mice, *Cyfip1*⁺/⁻ *n* = 8 mice and *Cyfip1*⁺/⁻ + cDNA *Cyfip1* *n* = 3 mice) (mean ± SEM; Two-way ANOVA with Tukey's multiple

comparisons test, $F_{(18, 144)}$ = 2.108, *p* = 0.0082; genotype effect *p* = 0.0594, bins effect *p* < 0.0001, interaction *p* = 0.0082; *\*p* < 0.05, *\*\*p* < 0.01). **b** Left, representative schematic and fluorescence image of a P15 brain slice after IUE at E15.5 with a tdTomato-expressing plasmid. Image preparation as in a. Scale bar 500 μm. Middle, representative fluorescence images of WT and *Cyfip1*⁺/⁻ mice contralateral cortex at postnatal day 15 (P15). Scale bar 100 μm. Right, quantification of axonal growth into the contralateral cortex as percentage of the total cortical thickness (WT *n* = 10 mice and *Cyfip1*⁺/⁻ *n* = 9 mice) (mean ± SEM; Two-tailed Mann-Whitney test, *p* = 0.0947). **c** Left, representative axonal arborization traces of WT and *Cyfip1*⁺/⁻ mice at P15 (main axonal branch, blue; primary branch, purple; secondary branch, green). Scale bar 50 μm. Right, quantification of contralateral axonal branches in WT and *Cyfip1*⁺/⁻ axons (WT *n* = 21 axons from 3 mice; *Cyfip1*⁺/⁻ *n* = 24 axons from 3 mice) (mean ± SEM; Two-tailed Multiple unpaired t-test, Holm-Sidak multiple comparisons test; primary *p* = 0.0015, secondary *p* = 0.00035 and total *p* = 0.00029). Source data are provided as a Source Data file.

mitochondria that we observed in *Cyfip1*⁺/⁻ axons. To test that, we loaded primary cortical neurons with the Ca²⁺ indicator Fluo-4 AM and measured the calcium concentration ([Ca²⁺]) in the axon (Fig. 3a) and the whole cell (Fig. S3a). We found that developing *Cyfip1*⁺/⁻ cortical neurons at DIV 3 exhibit significantly reduced cytoplasmic and axoplasmic [Ca²⁺] compared to WT (Fig. 3a and S3a).

Mitochondria uptake Ca²⁺ through ion channels present on the mitochondrial membrane. This process activates several enzymes involved in the mitochondrial electron transport chain, triggering ATP production[51]. Therefore, we argue that reduced intracellular Ca²⁺ availability may affect its uptake, leading to mitochondrial dysfunction/s. To test this hypothesis, we first expressed the calcium indicator

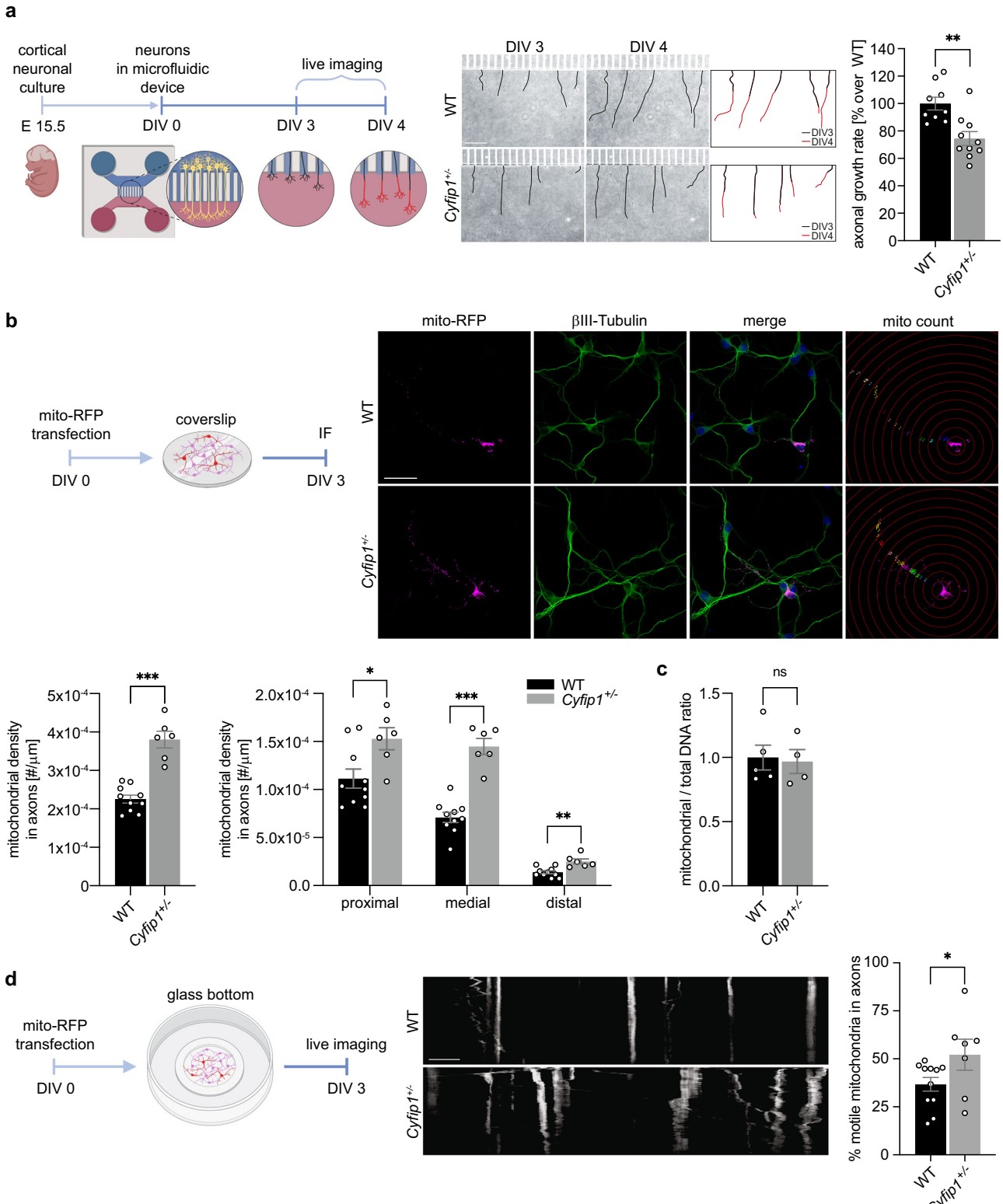

mitoGCaMP5G in DIV 3 primary cortical neurons and analyzed the calcium levels in the mitochondria. Our results revealed that mitochondria in *Cyfip1*[+/-] axons exhibit reduced calcium levels (Fig. 3b).

Next, to assess mitochondrial function, we measured mitochondrial membrane potential using TMRE, a fluorescent dye that selectively accumulates within mitochondria in a membrane potential-dependent manner. Our analysis revealed a reduction in mitochondrial membrane potential in both the axons (Fig. 3c) and cell bodies

(Fig. S3b) of *Cyfip1*[+/-] DIV 3 cortical neurons compared to WT. However, when evaluated at later developmental stages in vitro, this reduction was no longer observed (Fig. S3c, d). Finally, we further investigated mitochondrial function by measuring ATP levels and found that *Cyfip1*[+/-] cortical neurons exhibit reduced ATP compared to WT only at early developmental stages (DIV 3) (Fig. S3e, f, g).

To investigate the in vivo relevance of these findings, we measured axonal mitochondrial membrane potential in the corpus

**Fig. 2 | *Cyfip1*[+/-] cortical neurons exhibit reduced axonal growth, increased mitochondrial density and motility. a** Left, illustration depicting the experimental set up (created in BioRender). Middle, representative images of axons growing in the axonal compartment of microfluidic devices at day in vitro 3 (DIV 3) and DIV 4 for WT and *Cyfip1*[+/-] neurons. Scale bar 100 μm. Growth rate of each individual axon was quantified as represented in the scheme (black line represents DIV 3 and red line DIV 4). Right, average axonal growth rate in WT and *Cyfip1*[+/-] neurons (WT *n* = 9 embryos, 131 axons, and *Cyfip1*[+/-] *n* = 10 embryos, 170 axons) (mean ± SEM; Two-tailed Mann-Whitney test; *p* = 0.0021). **b** Upper left, illustration depicting the experimental set up (created in BioRender. https://BioRender.com/bko9ezm). Upper right, representative images of DIV 3 WT and *Cyfip1*[+/-] neurons after transfection with mito-RFP plasmid (magenta) and immunostaining for βIII-Tubulin (green). Mitochondrial density along the axon was calculated as represented in the mito count panel. Scale bar 20 μm. Bottom left, average mitochondrial density of the entire axon (WT *n* = 10 embryos, 37 axons, and *Cyfip1*[+/-] *n* = 6 embryos, 24 axons; mean ± SEM; Two-tailed Mann-Whitney test, *p* = 0.0002). Bottom right, average mitochondrial density for each axonal segment (proximal, medial, and distal) (WT *n* = 10 embryos, 37 axons, and *Cyfip1*[+/-] *n* = 6 embryos, 24 axons; mean ± SEM; Two-tailed Multiple Mann-Whitney test, "proximal" *p* = 0.0312, "medial" *p* = 0.0007, "distal" *p* = 0.0059). **c** Mitochondrial DNA copy number relative to nuclear DNA measured in WT and *Cyfip1*[+/-] neurons (WT *n* = 5 embryos, *Cyfip1*[+/-] *n* = 4 embryos; mean ± SEM; Two-tailed Mann-Whitney test, *p* = 0.7302). **d** Left, schematic representation of the experimental set up (created in BioRender). Middle, representative kymographs showing transport of mitochondria along axons of WT and *Cyfip1*[+/-] DIV 3 cortical neurons. Scale bar 10 μm. Right, percentage of motile mitochondria in WT and *Cyfip1*[+/-] DIV 3 cortical neurons (WT *n* = 11 embryos, 53 axons, *Cyfip1*[+/-] *n* = 7 embryos, 31 axons; mean ± SEM; Two-tailed Mann-Whitney test, *p* = 0.0414). Source data are provided as a Source Data file.

callosum of P5−6 mice and observed a reduction in *Cyfip1*[+/-] brain slices compared to WT (Fig. 3d). In addition, we observed that the ATP levels in the cortex of P5 mice are reduced in *Cyfip1*[+/-] compared with WT (Fig. S3h), while no difference was found at later stages (P15 and P30, Fig. S3i, j).

Changes in intracellular calcium concentration can also affect mitochondrial shape. Specifically, elevated cytosolic calcium levels can trigger increased mitochondrial fission or a mitochondrial shape transition, leading to a rounder mitochondrial shape[52,53]. This could in turn modulate mitochondrial function. To investigate if the observed reduction in cytosolic calcium in *Cyfip1*[+/-] neurons could influence mitochondrial shape, we performed electron microscopy on the axons of DIV 3 cortical neurons and examined their morphology. The analysis revealed a significantly increased mitochondrial area and elongation of the major axis of mitochondria in *Cyfip1*[+/-] axons. In contrast, the minor axis and roundness were comparable between WT and *Cyfip1*[+/-] axonal mitochondria (Fig. 3e). These results show that mitochondria appear larger and more elongated in *Cyfip1*[+/-] axons, aligning with reports linking calcium to mitochondrial shape[52,53].

Overall, our results suggest that CYFIP1 regulates mitochondrial function through the regulation of intracellular Ca²⁺ availability, which could critically impact axonal development in *Cyfip1*[+/-] cortical neurons.

## CYFIP1 regulates the expression of specific calcium channel subunits by interacting with Hu proteins

As shown by our laboratory and others, CYFIP1 regulates the translation of specific mRNAs[17,21,54]. A recently published study based on RNA immunoprecipitation sequencing (RIP-seq) experiments[54] suggested that some mRNAs encoding Ca²⁺ channel subunits are putative CYFIP1-targets, which could explain how CYFIP1 regulates intracellular [Ca²⁺]. To investigate whether the predicted Ca²⁺ channel subunits are regulated by CYFIP1, we immunoprecipitated CYFIP1 from DIV 3 primary cortical neurons and quantified the associated mRNAs by RT-qPCR. *mMap1b* mRNA, an mRNA previously identified in the CYFIP1 complex[21], was significantly associated with CYFIP1, >1.5 fold compared to *mHprt1* mRNA, which is not predicted to be in the CYFIP1 complex (Fig. 4a). Among the candidate mRNAs encoding protein subunits of Ca²⁺ channels, we found that *mCacna1c*, *mCacna1e* and *mCacna1i* mRNAs were significantly enriched, and therefore associated with the CYFIP1 complex. These mRNAs encode for the voltage-gated calcium channel (Ca$_V$) subunits alpha 1 C, 1E and 1I respectively (Fig. 4a). The alpha-1 subunit of Ca$_V$ channels is the pore-forming subunit and has a fundamental role in the selective gating and regulated entry of calcium ions into cells[55,56].

While CYFIP1 has no putative RNA binding domains, it was found in brain in a complex with several RNA binding proteins (RBP), such as FMRP, HuD and HuR[17]. Through the formation of these complexes, CYFIP1 could regulate several aspects of mRNA metabolism, such as translation, transport, or stability[17,57−60]. To understand how CYFIP1 regulates the alpha-1 subunits of Ca$_V$ channels, possibly leading to the observed defects in intracellular Ca²⁺ concentration, we first investigated the mRNA and protein levels of each subunit. By performing RT-qPCR from DIV 3 neurons, we found that the mRNA levels of two of the CYFIP1 targets, *mCacna1e* and *mCacna1i*, are significantly reduced in *Cyfip1*[+/-] compared to WT (Fig. 4b). The third target, *mCacna1c*, exhibits a notable inclination toward reduction but it did not reach statistical significance (*p* = 0.076626) (Fig. 4b) under the tested conditions.

The Ca$_V$ channel subunits are transmembrane proteins; therefore, to investigate the level of expression, we performed a Western Blot analysis on membrane-enriched fractions derived from DIV 3, DIV 7 and DIV 14 cortical neurons. We found that the protein levels of the three calcium channel targets, Ca$_V$1.2 (CACNA1C), Ca$_V$2.3 (CACNA1E) and Ca$_V$3.3 (CACNA1I), are significantly reduced in *Cyfip1*[+/-] DIV 3 neurons compared to WT (Fig. 4c). However, this reduction is normalized at later developmental stages (Fig. S4a). Conversely, the voltage-gated channel subunits gamma 2 (Ca$_V$γ2/CACNG2/Stargazin) and beta-3 (Ca$_V$β3/CACNB3) had comparable mRNA and protein levels between WT and *Cyfip1*[+/-] neurons at all developmental stages (Fig. 4b, c, Fig.S4a). This is consistent with Fig. 4a showing that the mRNA of those subunits does not co-IP with CYFIP1 and therefore those mRNAs are not part of the CYFIP1-complex.

Importantly, while at this early developmental time point (DIV 3), the identified Ca$_V$ channel subunits are distributed throughout the whole neuron, including cell body, dendrites, and axons (Fig. 4d−f), the protein levels of Ca$_V$2.3 and Ca$_V$3.3 are strikingly reduced in the axonal segment of *Cyfip1*[+/-] neurons compared to WT (Fig. 4e, f) suggesting a subcellular deficit already existing at this stage.

Together, these results demonstrate that CYFIP1 regulates the mRNA and protein levels of alpha-1 subunits of Ca$_V$ channels in developing cortical neurons and axons and suggest that the mechanism of regulation may be at the level of mRNA stability. As mentioned above, CYFIP1 regulates mRNA through its binding to specific RBPs such as FMRP, HuD and HuR among others[17]. While the CYFIP1-FMRP complex is a well-established regulator of translation initiation[17,21], Hu (nELAVL, neuronal ELAV-like) proteins are known to regulate the stability of their target mRNAs[61,62]. Moreover, HuD (nELAVL4) is known to be present in axons, and to be required for the axonal localization of certain mRNAs[63−65]. We hypothesize that the interaction between CYFIP1 and Hu proteins may play a role in modulating the stability of alpha-1 subunits of Ca$_V$ channels. In line with that, a published nELAVL CLIP-seq experiment from human brain found that *hCACNA1C, hCACNA1E* and *hCACNA1I* are targets of nELAVL proteins[66], and these mRNAs contain several RBP binding sites, especially on the 3' UTR, which suggest that nELAVL proteins could regulate their stability, localization, or translation (Fig. 5a and Fig. S4b).

To validate our hypothesis in the mouse model, we first used the RBPmap tool[67] to investigate the frequency of HuD (nELAVL4) and HuR (nELAVL1) binding sites on the mRNA of the calcium channel subunits (Fig. 5b and Fig. S4c). The prediction indicates that HuD and HuR have

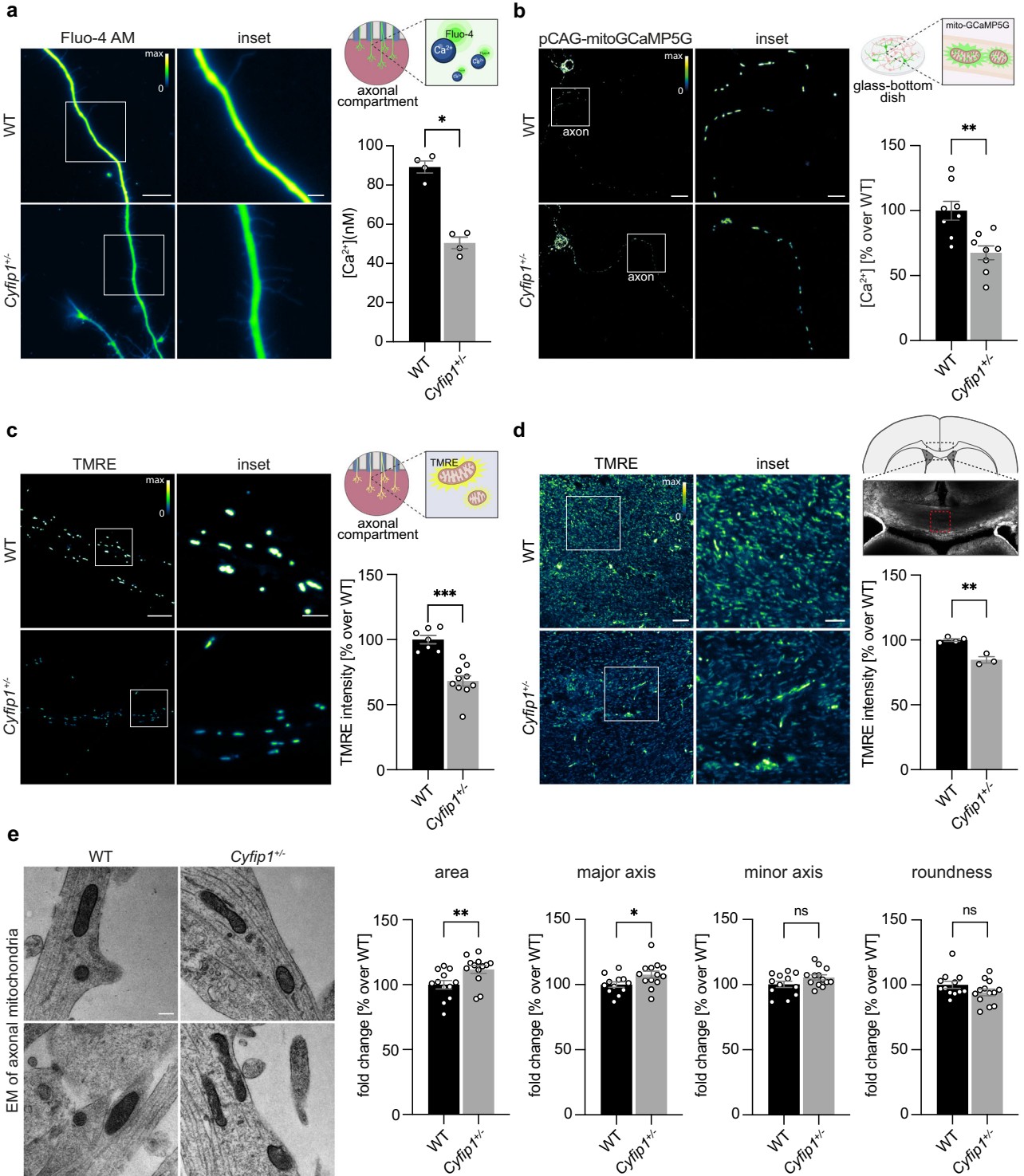

a high probability of binding to *mCacna1c*, *mCacna1e* and *mCacna1i* mRNAs, especially in their 3' UTR regions (Fig. 5b). Finally, alignments of the human and mouse sequences (5'UTR-coding-3'UTR) using BLAST (NCBI) show that the binding sites identified in human are conserved in mice, further supporting the potential interaction between HuD and HuR with *Cacna1c*, *Cacna1e*, and *Cacna1i* mRNAs in both species.

To further support our hypothesis, we first validated the interaction between CYFIP1 and Hu proteins in our model and specific developmental stage. We performed a CYFIP1 immunoprecipitation on protein extracts from DIV 3 WT cortical neurons and confirmed the

binding of CYFIP1 to HuD/Elavl4 and HuR/Elavl1 proteins (Fig. S4d), demonstrating the formation of this protein complex. However, CYFIP1 haploinsufficiency did not affect HuD/Elavl4 and HuR/Elavl1 protein levels (Fig. S4e). To assess the direct binding of Hu proteins to $Ca^{2+}$ channel subunit mRNAs, we immunoprecipitated HuD/Elavl4 and HuR/Elavl1 from DIV 3 WT and *Cyfip1*[+/-] cortical neurons and detected the associated mRNAs by RT-qPCR. The results revealed a significant reduction in the binding of *mCacna1c* mRNA to both HuD and HuR proteins in *Cyfip1*[+/-] neurons compared to WT. *mCacna1e* mRNA exhibited a significant reduction in its binding to HuD, and a trend in the association to HuR. In contrast, *mCacna1i* mRNA showed no

**Fig. 3 | CYFIP1 regulates intracellular Ca²⁺ levels and mitochondrial function.**
**a** Left, representative images of Fluo-4 AM intensity in the axons of WT and *Cyfip1*⁺/⁻ DIV 3 cortical neurons. Scale bar 10 µm, inset scale bar 2.5 µm. Up right, illustration depicting the experimental set up (created in BioRender. https://BioRender.com/sd45ds2). Bottom right, axoplasmic calcium concentration [Ca²⁺] (nM) measured using Fluo-4 AM imaging (WT *n* = 4 embryos, 31 axons, *Cyfip1*⁺/⁻ *n* = 4 embryos, 36 axons; mean ± SEM; Two-tailed Mann-Whitney test, *p* = 0.0286). **b** Left, representative images of mitoGCaMP5G fluorescence intensity in the axonal mitochondria of WT and *Cyfip1*⁺/⁻ DIV 3 and DIV 4 cortical neurons. Scale bar 20 µm. Inset scale bar 5 µm. Up right, illustration depicting the experimental set up (created in BioRender). Bottom right, histogram showing the axonal mitochondrial calcium concentration [Ca²⁺] expressed as a percentage over WT (WT *n* = 8 embryos, *Cyfip1*⁺/⁻ *n* = 8 embryos; mean ± SEM; Two-tailed Mann-Whitney test, *p* = 0.0030). **c** Left, representative images of TMRE intensity in axonal mitochondria of WT and *Cyfip1*⁺/⁻ DIV 4 neurons. Scale bar 10 µm, inset scale bar 2.5 µm. Up right, illustration depicting the experimental set up (created in BioRender). Bottom right, average

TMRE intensity in WT and *Cyfip1*⁺/⁻ axonal mitochondria, expressed as a percentage over WT (WT *n* = 7 embryos, 35 axons, *Cyfip1*⁺/⁻ *n* = 10 embryos, 50 axons; mean ± SEM; Two-tailed Mann-Whitney test, *p* = 0.0001). **d** Left, representative images showing TMRE intensity in the corpus callosum of acute organotypic brain slices from WT and *Cyfip1*⁺/⁻ P5-6 mice. Scale bar 10 µm. Inset scale bar 5 µm. Up right, illustration depicting the brain region analyzed. Bottom right, histograms showing TMRE intensity expressed as a percentage over WT. (WT *n* = 4 animals, 14 organotypic brain slices, *Cyfip1*⁺/⁻ *n* = 3, 10 organotypic brain slices. mean ± SEM; Two-tailed Unpaired *t*-test, *p* = 0.0015). **e** Left, representative electron microscopy (EM) images of mitochondria in WT and *Cyfip1*⁺/⁻ DIV 3 cortical axons. Scale bar 250 nm. Right, histograms showing average mitochondrial area, major axis, minor axis and roundness, expressed as percentage over WT. (WT *n* = 12, *Cyfip1*⁺/⁻ *n* = 13; mean ± SEM; Two-tailed Mann–Whitney test; area *p* = 0.0055, major axis *p* = 0.0398, minor axis *p* = 0.2101, roundness *p* = 0.1519). Source data are provided as a Source Data file.

---

statistically significant difference, potentially due to a high variability at this developmental stage or regulation independent of HuD/Elavl4 or HuR/Elavl1 (Fig. 5c, d).

Together, these findings prompted us to investigate if Hu proteins could regulate *mCacna1c, mCacna1e* and *mCacna1i* mRNAs at the level of stability. We treated primary cortical neurons with the transcriptional inhibitor Actinomycin D and measured the respective mRNA levels during a time course (0, 2 and 4 h after treatment). Such a treatment revealed that the mRNA decay of the three calcium channel targets, *mCacna1c, mCacna1e* and *mCacna1i*, is enhanced in *Cyfip1*⁺/⁻ neurons after 4 h (Fig. 5e). The highly stable *mHprt1* mRNA exhibited a prolonged half-life, while *mArc* mRNA, a transcript that is activity-regulated and described as unstable[68–70], shows a great decay in both genotypes starting from 2 h of actinomycin-D treatment (Fig. S4f). The non-target *mCacng2* and *mCacnb3* mRNAs showed no difference in stability between the two genotypes (Fig. S4g).

Altogether, our results demonstrate that, during cortical axonal development, CYFIP1 regulates the axonal protein levels of alpha-1 subunits of Ca$_V$ channels by controlling their mRNA stability through its binding with Hu proteins. Based on these data, we hypothesize that a reduced expression of these subunits could lead to decreased intracellular calcium concentrations, thereby impacting physiological processes in neurons, and having major consequences for normal brain physiology and function. Our findings substantiate the hypothesis that, at this very early developmental stage, the regulatory influence of CYFIP1 on intracellular Ca²⁺ levels may underlie the aberrant modulation of mitochondrial function and the observed disruptions in axonal development found in *Cyfip1*⁺/⁻ cortical neurons.

### Elevating intracellular Ca²⁺ concentration mitigates the observed axonal and mitochondrial phenotypes

Employing microfluidic devices, we examined whether a 24 h treatment with Ionomycin, augmenting intracellular [Ca²⁺], could rectify the previously observed delayed axonal growth in *Cyfip1*⁺/⁻ cortical neurons.

We therefore used Ionomycin, an ionophore that binds calcium and increases its transport across the plasma membrane, ultimately increasing intracellular [Ca²⁺][71,72]. As a control, we treated the neurons with BAPTA-AM, chelator with a high selectivity for Ca²⁺ that accumulates in the cell and sequesters calcium (Fig. S5a).

While Ionomycin significantly increased axonal growth in both WT and *Cyfip1*⁺/⁻ neurons (Fig. 6a), the growth rate became comparable in both genotypes, demonstrating that increasing intracellular [Ca²⁺] successfully restored axonal growth (Fig. 6a). BAPTA-AM treatment markedly decreased the growth rate in WT neurons and only marginally affected *Cyfip1*⁺/⁻ neurons, where the growth rate was already low (Fig. S5b).

We then asked if Ionomycin treatment could also restore mitochondrial motility and function in *Cyfip1*⁺/⁻ axons. We treated cortical neurons with Ionomycin for 24 h and analyzed the percentage of motile mitochondria and their membrane potential as described above. We found that, while Ionomycin treatment did not affect mitochondrial motility in WT, it was able to reduce the excess motility in *Cyfip1*⁺/⁻ axons to a level comparable to the WT, therefore completely rescuing the phenotype (Fig. 6b). Similarly, Ionomycin did not affect mitochondrial membrane potential (measured by TMRE intensity) in WT but was able to increase it in *Cyfip1*⁺/⁻ axons to a level comparable to the WT, leading, also in this case, to a complete rescue of the phenotype (Fig. 6c).

Finally, to validate the direct role of the identified calcium channels in driving the observed axonal growth phenotype, we treated DIV3 neurons with the voltage-gated calcium channels (VGCCs) agonists Bay-K-8644 (50 µM)[73] and Nefiracetam (10 µM)[74] and monitored their growth over 24 h. Bay-K-8644 specifically activates L-type calcium channels, whereas Nefiracetam has a broader effect, targeting both L- and N-type calcium channels. CACNA1C, the most affected channel in *Cyfip1*⁺/⁻ cortical neurons, is an L-type calcium channel, and is therefore activated by both Bay-K-8644 and Nefiracetam. Our results demonstrate that treatment with both agonists effectively rescues the delayed axonal growth phenotype (Fig. 6d), validating the direct role of calcium channels in the axonal growth phenotypes observed in *Cyfip1*⁺/⁻ neurons.

Altogether, these results demonstrate that increasing intracellular [Ca²⁺] restores the mitochondrial and axonal phenotypes observed in *Cyfip1*⁺/⁻ neurons, suggesting that appropriate calcium levels play a crucial role in early axonal development.

## Discussion

Here we found a crucial role for the disease-associated protein CYFIP1 in cortical axonal development. We discovered that a reduction in CYFIP1, which mimics the psychiatric disorders seen with 15q11.2 copy number deletion, leads to reduced axonal growth, intracellular calcium concentration, and mitochondrial function. At the molecular level, we demonstrated that CYFIP1 regulates the membrane expression of voltage-gated calcium channels by controlling the stability of their mRNAs. Notably, upregulation of intracellular calcium was sufficient to rescue the delayed axonal growth and mitochondrial dysfunction observed in *Cyfip1*⁺/⁻ neurons. Altogether, our findings provide cellular and molecular evidence for the role of the disease-associated protein CYFIP1 in cortical axonal development (Fig. 7).

Cytosolic calcium plays a critical role in controlling axonal and dendritic growth and arborization[75–79]. While some studies have reported that a sudden calcium increase in growth cones negatively regulates axon extension[80–83], others have found that elevating [Ca²⁺] promotes outgrowth[84]. Furthermore, calcium waves play a crucial role

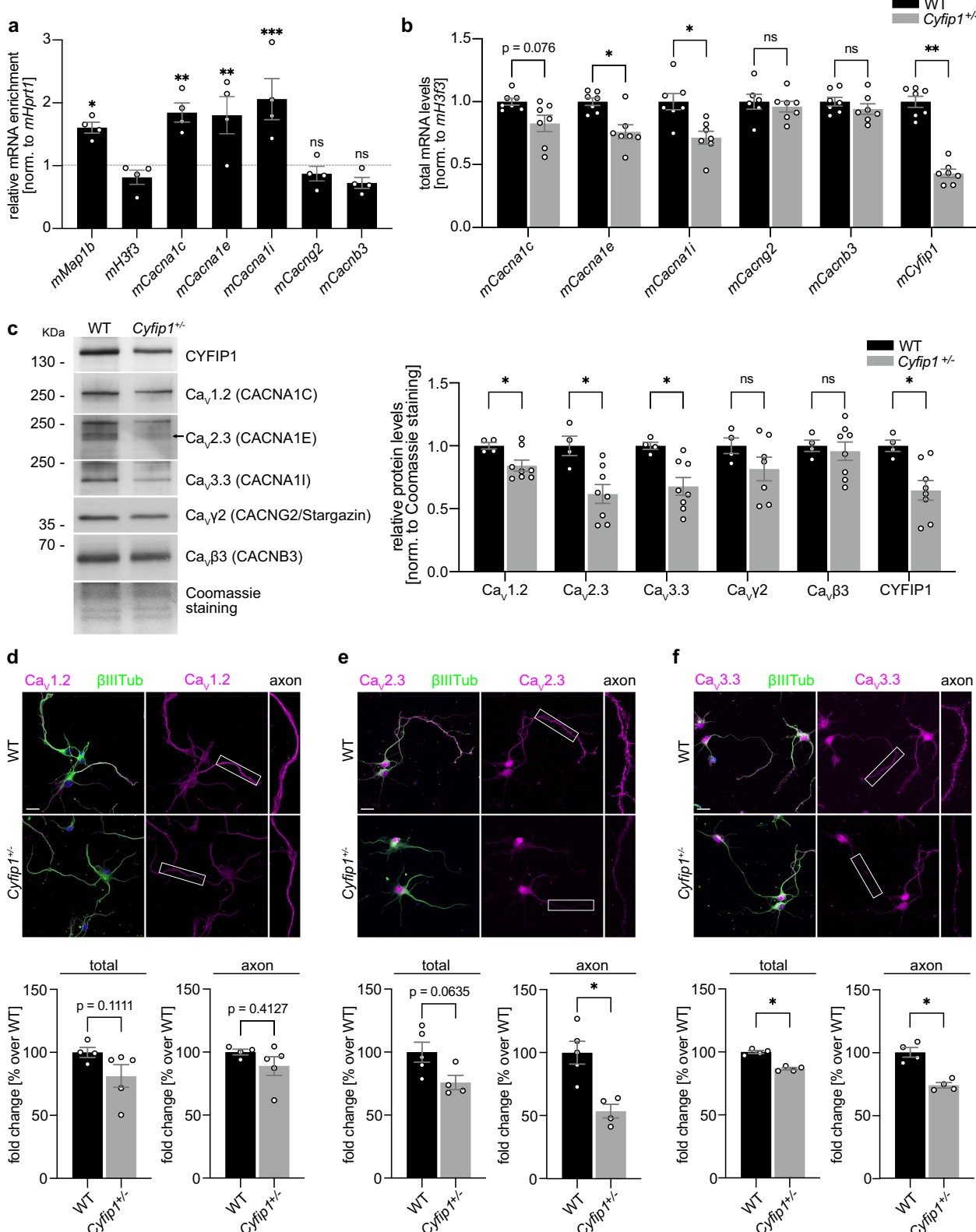

in axonal signaling. When neurons fire, calcium ions surge into the axon, facilitating neurotransmitter release at the synapse. This process ensures efficient communication between neurons and influences axonal extension, migration, and the positioning of the neurons in the correct cortical layer[85,86]. Moreover, different signals, and the local distribution and magnitude of the calcium transients along the axon can lead to diverse outcomes. For instance, in cortical neurons cultured in vitro, axonal processes with higher frequencies of calcium transients will extend at the expense of other processes with lower calcium activity, which will retract[87].

Cytosolic $Ca^{2+}$ fluctuations also play a pivotal role in maintaining the structure and function of intracellular organelles[88,89]. Notably, cytosolic calcium has been implicated in modulating mitochondrial morphology, promoting transitions between tubular and spherical

**Fig. 4 | CYFIP1 regulates the expression of calcium channels subunits. a** CYFIP1 RNA immunoprecipitation (RNA-IP) from DIV 3 WT cortical neurons. Histogram showing relative enrichment of the mRNAs over the non-specific IgG, measured by RT-qPCR of the eluate. The values were normalized for the input and *mHprt1* mRNA and expressed as fold change over the non-specific IgG of each mRNA ($n = 4$ embryos; mean ± SEM; One-Way ANOVA $p < 0.0001$; *mMap1b* mRNA $p = 0.0390$, *mCacna1c* mRNA $p = 0.0054$, *mCacna1e* mRNA $p = 0.0078$, *mCacna1i* mRNA $p = 0.0009$, *mCacng2* mRNA $p = 0.9997$, *mCacnb3* mRNA $p = 0.9983$). **b** Total mRNA levels of the $Ca^{2+}$ channels in DIV 3 WT and *Cyfip1*[+/-] cortical neurons. Histograms represent *mCacna1c*, *mCacna1e*, *mCacna1i*, *mCacng2*, *mCacnb3* and *mCyfip1* mRNA levels, normalized to *mH3f3* levels and expressed as a fold change over WT (WT $n = 6/7$ embryos, *Cyfip1*[+/-] $n = 7$ embryos; mean ± SEM; Two-tailed Multiple Mann-Whitney test, *mCacna1c* mRNA $p = 0.0766$, *mCacna1e* mRNA $p = 0.0435$, *mCacna1i* mRNA $p = 0.0202$, *mCacng2* mRNA $p = 0.6282$, *mCacnb3* mRNA $p = 0.5343$, *mCyfip1* mRNA $p = 0.0034$). **c** Left, representative Western Blot showing CYFIP1, $Ca_V1.2$ (CACNA1C), $Ca_V2.3$ (CACNA1E), $Ca_V3.3$ (CACNA1I), $Ca_V\gamma2$ (CACNG2/Stargazin) and $Ca_V\beta3$ (CACNB3) in membrane-enriched fractions from WT and *Cyfip1*[+/-] DIV 3 cortical neurons. The molecular weight of each protein is indicated in kDa. Right, histogram representing $Ca_V1.2$, $Ca_V2.3$, $Ca_V3.3$, $Ca_V\gamma2$, $Ca_V\beta3$ and CYFIP1 protein expression levels in membrane-enriched fractions from WT and *Cyfip1*[+/-] DIV 3 cortical neurons. Protein levels were normalized to Coomassie staining (WT $n = 4$ embryos, *Cyfip1*[+/-] $n = 7/8$ embryos; mean ± SEM; Two-tailed Multiple unpaired $t$-test, $Ca_V1.2$ $p = 0.0338$, $Ca_V2.3$ $p = 0.0281$, $Ca_V3.3$ $p = 0.0129$, $Ca_V\gamma2$ $p = 0.2574$, $Ca_V\beta3$ $p = 0.6259$, CYFIP1 $p = 0.0137$). **d–f** Representative images from WT and *Cyfip1*[+/-] DIV 3 cortical neurons stained for $Ca_V1.2$, $Ca_V2.3$, $Ca_V3.3$ (magenta) and βIII-Tubulin (green) (scale bar 20 μm). Histograms show the fluorescence intensity of each calcium channel normalized to βIII-Tubulin in the total neuron (left) and in the axon (right), expressed as a percentage over WT ($Ca_V1.2$: WT $n = 4$ embryos, *Cyfip1*[+/-] $n = 5$ embryos; mean ± SEM; Two-tailed Mann-Whitney test, total $p = 0.1111$, axon $p = 0.4127$; $Ca_V2.3$: WT $n = 5$ embryos, *Cyfip1*[+/-] $n = 4$ embryos; mean ± SEM; Two-tailed Mann-Whitney test, total $p = 0.0635$, axon $p = 0.0159$; $Ca_V3.3$: WT $n = 4$ embryos, *Cyfip1*[+/-] $n = 4$ embryos; mean ± SEM; Two-tailed Mann-Whitney test, total $p = 0.0286$, axon $p = 0.0286$). Source data are provided as a Source Data file.

shapes[53,90]. In this study, we demonstrate that *Cyfip1* haploinsufficiency in neurons leads to reduced axoplasmic calcium levels, potentially contributing to the increased mitochondrial area and length observed in the heterozygous model.

Calcium enters neurons primarily through voltage-sensitive calcium ($Ca_V$) channels, ionotropic glutamate receptors, and transient receptor potential channels[79,91]. $Ca_V$ ion channels translate neuronal activity into rapid intracellular calcium signals, triggering diverse cellular responses. Their relevance to major neurological and psychiatric disorders including ASD, SCZ, and bipolar disorder[10,92–94], along with their potential as therapeutic targets, has driven research into the subcellular mechanisms modulated by $Ca_V$ channel activity.

Alterations in the levels of these channels compromise calcium entry, thereby affecting cellular development and function. Consistently, previous studies have shown that the expression of $Ca_V1.2$ or $Ca_V1.3$ (subunits of the L-type voltage-gated calcium channel) is sufficient to trigger spontaneous regenerative calcium transients in developing cortical neurons, while knocking out $Ca_V1.2$ reduces neurite length[95]. Furthermore, embryonic deletion of $Ca_V1.2$ in cortical neurons disturbs spontaneous calcium activity, causing increased anxiety in mice[96], whereas depletion of intracellular calcium in rat cortical neurons arrests axonal growth and reduces dendritic arborization[97]. These findings support our model in which CYFIP1 regulation of calcium channel subunits compromise intracellular calcium levels, resulting in delayed axonal development[10,84,85] (Fig. 7).

Exploring the dynamic interplay between RNA-binding proteins and calcium homeostasis and signalling–crucial for maintaining cellular functions and responding to environmental cues–opens a novel frontier in neuroscience. Currently, the only evidence relates to two specific RNA-binding proteins regulating splicing[98]. Specifically, RNA-binding Fox (Rbfox) proteins associate with $Ca_V2.2$, regulating the splicing of specific exons in peripheral neurons[99], while the splicing factor Nova controls alternative splicing in particular subtypes of $Ca_V2$ calcium channels[100]. However, the impact of such differential splicing on axonal growth and brain wiring has not been addressed.

Here, using CYFIP1 heterozygosity as a model, we demonstrate that mRNAs encoding for the alpha-1 subunits of $Ca_V$ channels are regulated at the level of stability, specifically while the axon is developing, by the ribonucleocomplex constituted by CYFIP1 and the Hu proteins (Fig. 5, Fig. 6 and Fig.S4).

Dysregulation of voltage-gated calcium ($Ca_V$) channels can lead to various phenotypes within cells, and the specific outcome depends on the cell type, tissue, and context of the dysregulation. This underscores the critical role of $Ca_V$ channels in maintaining cellular homeostasis and function. Relevant to our work, excessive or prolonged activation of $Ca_V$ channels leads to calcium overload in mitochondria, disrupting their normal function at multiple levels and significantly impacting cellular development and homeostasis[101,102]. On one side, $Ca^{2+}$ activates mitochondrial metabolism and stimulates ATP synthesis, on the other side mitochondria also buffer cytosolic calcium[51,103,104]. Several studies over the last years suggest that a proper balance between motile and stationary mitochondria is crucial for maintaining the high ATP levels required for proper axonal growth and branching[105,106]. Moreover, it has been largely described that $Ca^{2+}$ is a key regulator of mitochondria motility and docking[44–49,107]. Here, we found that *Cyfip1*-deficient axons have increased mitochondrial density and transport, but reduced calcium concentration and activity (Figs. 2 and 3). Our hypothesis posits that the diminished intracellular calcium levels observed in *Cyfip1*[+/-] neurons may adversely impact both mitochondrial transport and activity, potentially contributing to the observed delayed axonal growth phenotype in these neurons. Remarkably, Ionomycin treatment in *Cyfip1*[+/-] neurons was able to rescue not only the delayed axonal growth but also the increased motility and reduced mitochondrial activity (Fig. 6). Additionally, Bay-K-8644 and Nefiracetam, two agonists of L-type calcium channels, fully rescue the delayed axonal growth in *Cyfip1*[+/-] neurons, further demonstrating the role of these calcium channels in the observed phenotype. Altogether, we propose that CYFIP1 regulation of voltage gated calcium channels, leading to defects in intracellular calcium levels, may impact several aspects of axonal function, among them mitochondrial distribution and activity, ultimately compromising axonal development.

The development of callosal projections is tightly regulated. The axons of layer II/III callosal projection neurons (CPNs) in the somatosensory cortex cross the midline at postnatal day 3 (P3) and start invading their homotopic contralateral region at P5-P6. By P7, callosal axons reach the superficial layers of the contralateral cortex and undergo extensive arborization during the second postnatal week[36], being later refined up to postnatal stage 21[37]. The delayed growth and arborization observed in *Cyfip1*[+/-] axons (Fig. 1) may impair target selection, ultimately affecting synaptic transmission and brain connectivity. Indeed, previous work from our laboratory found that *Cyfip1*[+/-] mice have reduced bilateral functional connectivity and structural abnormalities in the corpus callosum[28]. Similar observation using diffusion tensor imaging, showed extensive white matter changes in *Cyfip1*[+/-] rats, which were most pronounced in the corpus callosum[32]. Consistent with this idea, Mukai and colleagues, using a mouse model with 22q11.2 deletion, specifically designed for studying SCZ, found that deficits in callosal axonal development and arborization compromise synaptic transmission of adult mice[108], which also present altered functional connectivity[109].

Moreover, brain connectivity mapping across 16 mouse models for autism reveals a spectrum of functional connectivity abnormalities across different etiologies[110]. This highlights connectivity defects as a

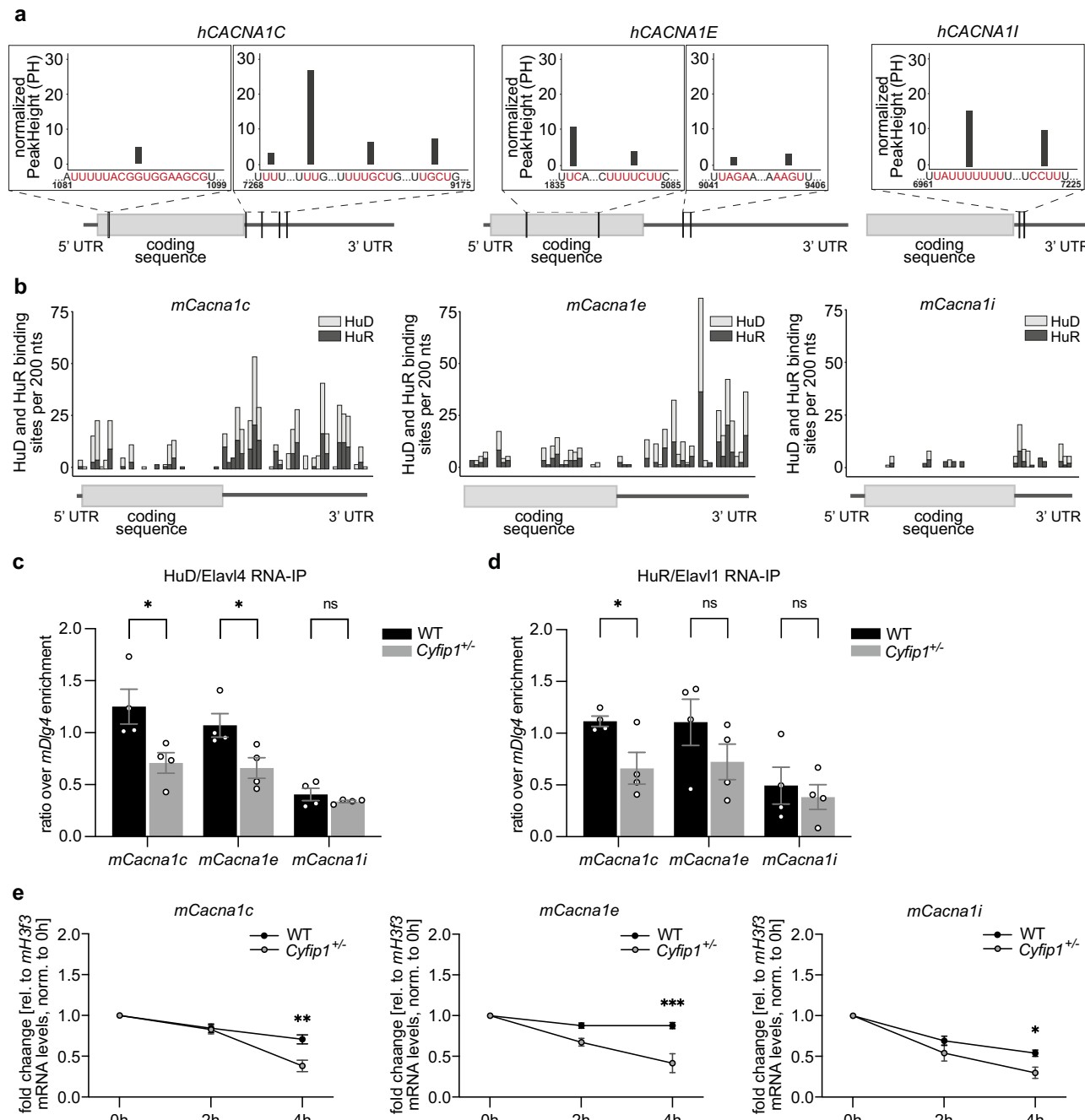

**Fig. 5 | Multiple sites for nELAVL proteins are present on the mRNAs encoding the calcium channel subunits. a** Histograms indicating the frequency of nELAVL binding sites (represented as normalized Peak Height) on the coding sequence and 3′ untranslated region (UTR) of human *hCACNA1C*, *hCACNA1E* and *hCACNA1I* mRNAs, as found in previous CLIP-seq published data[66]. **b** Histograms representing the frequency of predicted binding sites of HuD and HuR on the 5′ UTR, coding sequence and 3′UTR of mouse *mCacna1c*, *mCacna1e* and *mCacna1i* mRNAs, generated using RBPmap[67]. **c** HuD/Elavl4 RNA immunoprecipitation (RNA-IP) from DIV 3 WT and *Cyfip1*[+/-] cortical neurons. Histograms represent calcium channel mRNAs enrichment, calculated as the ratio over the positive control (*mDlg4*) (WT n = 4 embryos, *Cyfip1*[+/-] n = 4 embryos; mean ± SEM, Two-tailed Multiple Mann-Whitney test; *mCacna1c* mRNA p = 0.0285, *mCacna1e* mRNA p = 0.0285, *mCacna1i* mRNA p = 0.8857). **d** HuR/Elavl1 RNA-IP from DIV 3 WT and *Cyfip1*[+/-] cortical neurons. Histograms represent calcium channel mRNAs enrichment, calculated as the ratio over the positive control (*mDlg4*) (mean ± SEM, Two-tailed Multiple Mann-Whitney

test; *mCacna1c* mRNA p = 0.0310, *mCacna1e* mRNA p = 0.2270, *mCacna1i* mRNA p = 0.6289). **e** Histograms representing the mRNA decay after transcriptional shutdown with Actinomycin D in WT and *Cyfip1*[+/-] DIV 3 cortical neurons. mRNA expression levels of each calcium channel subunit *mCacna1c*, *mCacna1e* and *mCacna1i* were normalized to *mH3f3* mRNA at each time point (0, 2, and 4 h after Actinomycin D treatment (*mCacna1c*: WT n = 3 embryos, *Cyfip1*[+/-] n = 4 embryos; Two-way ANOVA, $F_{(2, 15)} = 6.834$, p = 0.0078; time p < 0.0001, genotype p = 0.0137, interaction p = 0.0078; Sidak's multiple comparisons test, 2 h p = 0.9957, 4 h p = 0.0010; *mCacna1e*: WT n = 3 embryos, *Cyfip1*[+/-] n = 4 embryos; Two-way ANOVA, $F_{(2, 15)} = 6.697$, p = 0.0083; time p = 0.0002, genotype p = 0.0006, interaction p = 0.0083; Sidak's multiple comparisons test, 2 h p = 0.1052, 4 h p = 0.0003; *mCacna1i*: WT n = 3 embryos, *Cyfip1*[+/-] n = 4 embryos; mean ± SEM, Two-way ANOVA, $F_{(2, 15)} = 1.905$, p = 0.1831; time p < 0.0001, genotype p = 0.0218, interaction p = 0.1831; Sidak's multiple comparisons test, 2 h p = 0.2958, 4 h p = 0.0454). Source data are provided as a Source Data file.

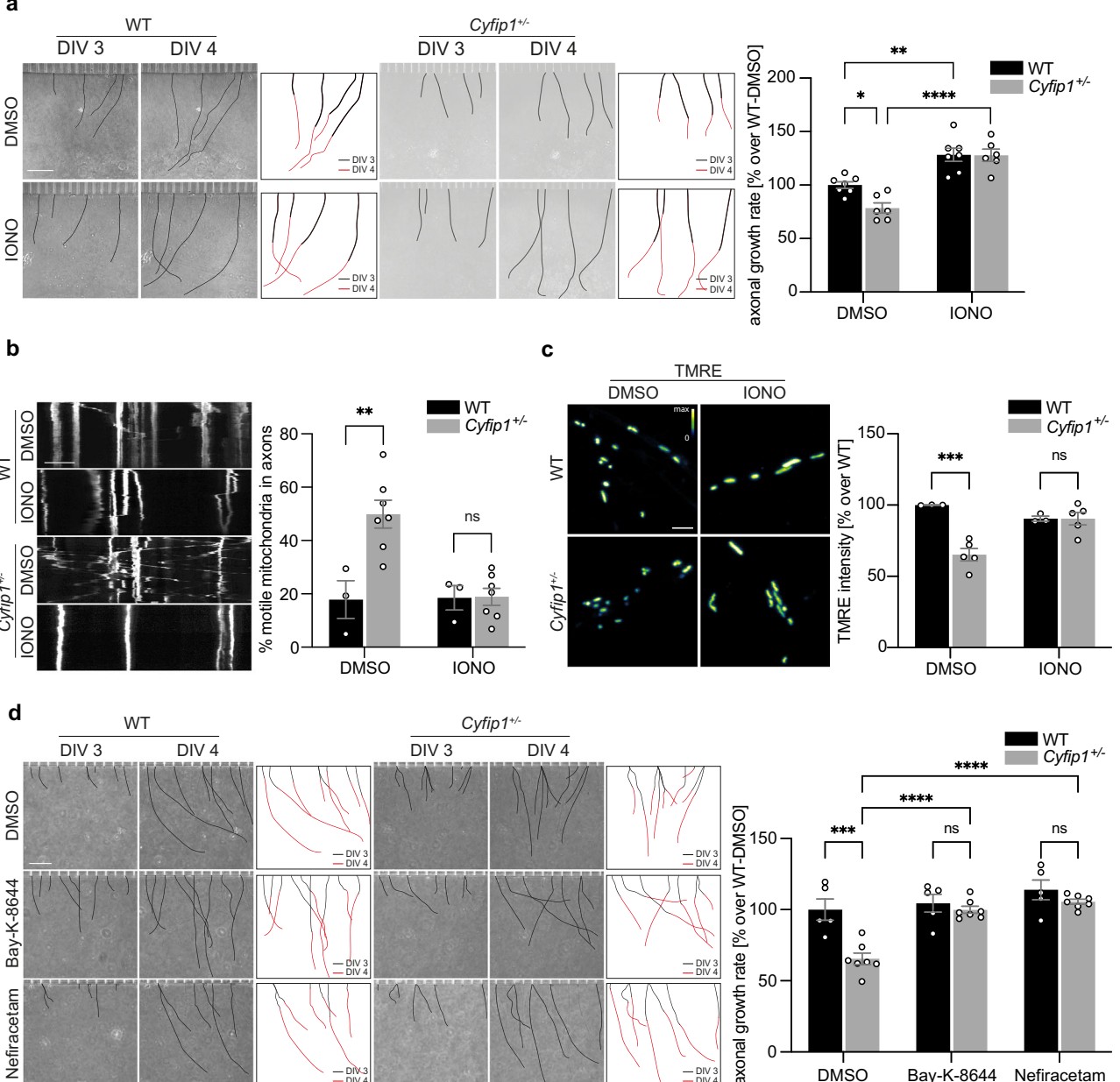

**Fig. 6 | Elevated intracellular Ca²⁺ levels rescue the axonal and mitochondrial phenotypes in *Cyfip1*⁺ᐟ⁻ neurons. a** Left, representative images of axons growing in the axonal compartment of microfluidic devices at day in vitro 3 (DIV 3) and DIV 4 for WT and *Cyfip1*⁺ᐟ⁻ neurons treated with DMSO or Ionomycin (IONO, 1 μM). Scale bar 200 μm. Axonal growth rate was quantified (black line, DIV 3; red line, DIV 4). Right, average axonal growth rate in WT and *Cyfip1*⁺ᐟ⁻ neurons treated with DMSO or Ionomycin, expressed as a percentage over WT-DMSO (WT $n = 7$ embryos, 149 DMSO-treated axons and 143 Ionomycin-treated axons; *Cyfip1*⁺ᐟ⁻ $n = 6$ embryos, 121 DMSO-treated axons and 148 Ionomycin-treated axons; mean ± SEM; Two-Way ANOVA, $F_{(1,22)} = 4.326$, $p = 0.0494$; genotype effect $p < 0.0001$, treatment effect $p = 0.0420$, interaction $p = 0.0494$; Sidak's multiple comparisons test, WT-DMSO vs. *Cyfip1*⁺ᐟ⁻-DMSO $p = 0.0392$, *Cyfip1*⁺ᐟ⁻-DMSO vs. *Cyfip1*⁺ᐟ⁻-IONO $p < 0.0001$, WT-DMSO vs. WT-IONO $p = 0.0027$). **b** Left, representative kymographs showing mitochondrial transport along axons of WT and *Cyfip1*⁺ᐟ⁻ DIV 4 cortical neurons treated with DMSO or Ionomycin (1 μM). Scale bar 10 μm. Right, histogram shows percentage of motile mitochondria in WT and *Cyfip1*⁺ᐟ⁻ axons treated with DMSO or Ionomycin (WT $n = 3$ embryos, 195 DMSO-treated mitochondria and 157 Ionomycin-treated mitochondria; *Cyfip1*⁺ᐟ⁻ $n = 7$ embryos, 393 DMSO-treated mitochondria and 398 Ionomycin-treated mitochondria; mean ± SEM; Two-Way ANOVA, $F_{(1,16)} = 8.386$, $p = 0.0105$; treatment effect $p = 0.0134$, genotype effect $p = 0.0091$,

interaction $p = 0.0105$; WT-DMSO vs. *Cyfip1*⁺ᐟ⁻-DMSO $p = 0.0045$, WT-IONO vs. *Cyfip1*⁺ᐟ⁻-IONO p > 0.9999). **c** Left, representative images of TMRE intensity in axonal mitochondria of WT and *Cyfip1*⁺ᐟ⁻ neurons treated with DMSO or Ionomycin (1 μM). Scale bar 2.5 μm. Right, average TMRE intensity expressed as a percentage over WT-DMSO (WT $n = 3$ embryos, *Cyfip1*⁺ᐟ⁻ $n = 5$ embryos; mean ± SEM; Two-Way ANOVA, $F_{(1,12)} = 16.87$, $p = 0.0015$; treatment effect $p = 0.0866$, genotype effect $p = 0.0014$, interaction $p = 0.0015$; WT-DMSO vs. *Cyfip1*⁺ᐟ⁻-DMSO $p = 0.0004$, WT-IONO vs. *Cyfip1*⁺ᐟ⁻-IONO p > 0.9999). **d** Left, representative images of axons at DIV 3 and DIV 4 for WT and *Cyfip1*⁺ᐟ⁻ neurons treated with DMSO, (S)-(-)-Bay-K-8644 (50 μM) or Nefiracetam (10 μM). Scale bar 200 μm. Right, average axonal growth rate expressed as a percentage over WT-DMSO (WT $n = 5$ embryos, 142 DMSO-treated axons, 190 Bay-K-8644-treated axons and 189 Nefiracetam-treated axons; *Cyfip1*⁺ᐟ⁻ $n = 7$ embryos, 150 DMSO-treated axons, 258 Bay-K-8644-treated axons and 207 Nefiracetam-treated axons; mean ± SEM; Two-Way ANOVA, $F_{(2,30)} = 5.8$, $p = 0.0070$; genotype effect $p = 0.0003$, treatment effect $p < 0.0001$, interaction $p = 0.0070$; Sidak's multiple comparisons test, WT-DMSO vs. *Cyfip1*⁺ᐟ⁻-DMSO $p = 0.0002$, *Cyfip1*⁺ᐟ⁻-DMSO vs. *Cyfip1*⁺ᐟ⁻-Bay-K-8644 $p < 0.0001$, *Cyfip1*⁺ᐟ⁻-DMSO vs. *Cyfip1*⁺ᐟ⁻-Nefiracetam $p < 0.0001$, WT-Bay-K-8644 vs. *Cyfip1*⁺ᐟ⁻-Bay-K-8644 $p = 0.9824$, WT-Nefiracetam vs. *Cyfip1*⁺ᐟ⁻-Nefiracetam $p = 0.8057$). Source data are provided as a Source Data file.

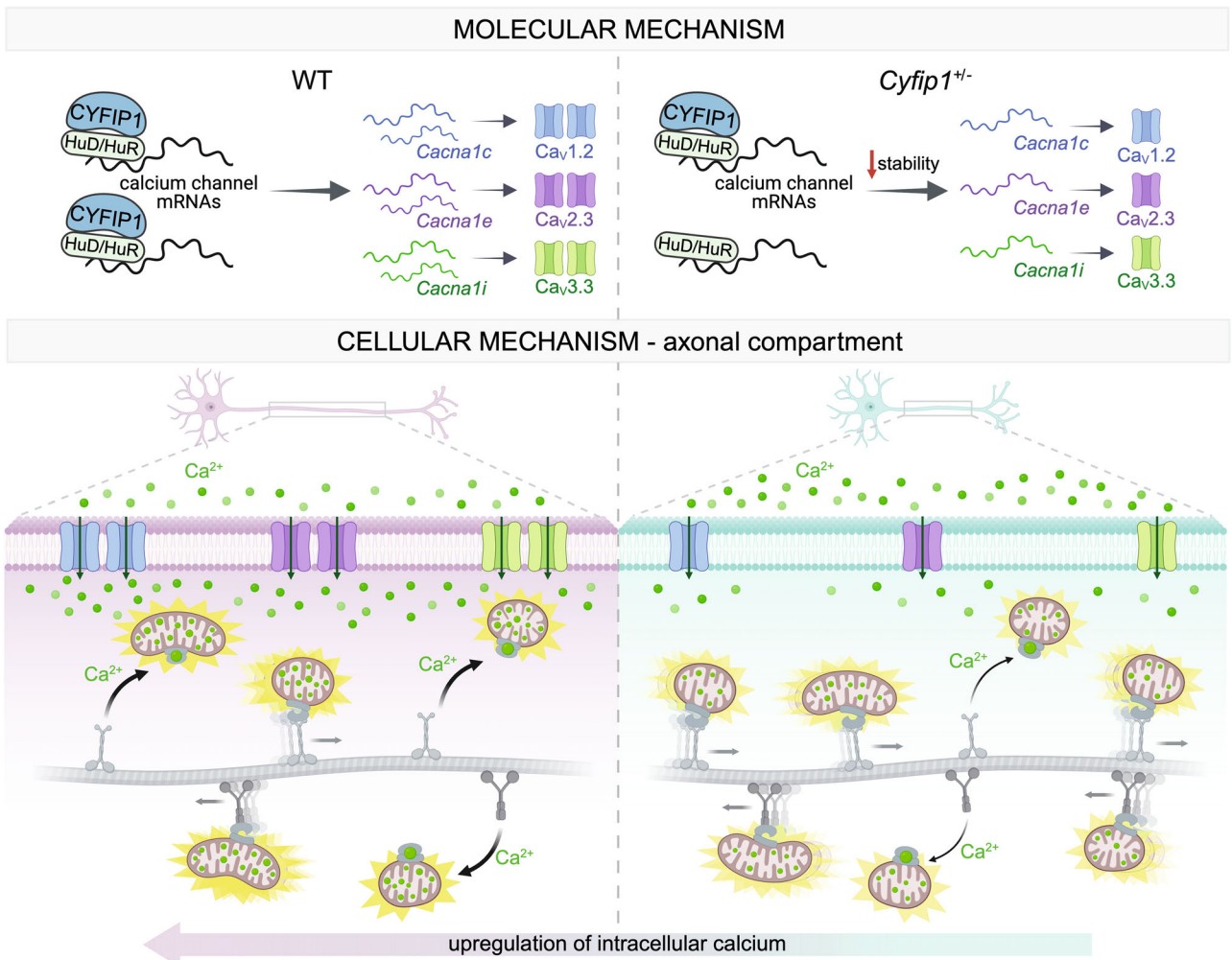

**Fig. 7 | Illustrative model depicting the proposed regulation of cortical axonal growth by CYFIP1.** CYFIP1, potentially interacting with the RNA binding proteins HuD and HuR (previously identified as CYFIP1 interactors), is implicated in regulating the mRNA stability of calcium channel subunits *Cacna1c*, *Cacna1e* and *Cacna1i*. In *Cyfip1*[+/-] neurons, reduced CYFIP1 levels result in a decrease protein abundance of the regulated calcium channel subunits, consequently leading to a decrease in intracellular and mitochondrial calcium concentration. Low levels of

calcium ions may affect mitochondria polarity and motility, both of which we found impaired in *Cyfip1*[+/-] axons. The decreased calcium concentration and the mitochondrial defects concur in reducing axonal growth observed in *Cyfip1*[+/-] neurons. By restoring the intracellular calcium homeostasis, both the axonal growth and mitochondrial defects are rescued. Created in BioRender. https://BioRender.com/xquv8cy.

common abnormality of ASD, which could potentially be used to stratify individuals with different aetiology. Finally, a wealth of evidence links abnormalities in the corpus callosum and interhemispheric connectivity (driven by callosal projection neurons) to many neurodevelopmental disorders, among them ASD and SCZ[2,3,111,112]. Therefore, gaining deeper insights into the molecular mechanisms regulating cortical development and brain connectivity in neurodevelopmental disorders is crucial. This understanding may pave the way for future therapeutic strategies, especially given the current lack of a cure for ASD[113].

Our results further support the crucial role of CYFIP1 in neurodevelopmental disorders, especially those marked by disturbed brain connectivity and wiring, and provide insights into the molecular mechanisms by which CYFIP1 may regulate axonal development, ultimately contributing to brain connectivity.

## Methods

### Mouse strains

Animal housing and care was conducted according to the institutional guidelines that are in compliance with national and international laws and policies (European Directive 2010/63/EU on the protection of

animals used for scientific purposes of 20 October 2010, Italian D.Lgs 26/2014, Swiss Loi fédérale sur la protection des animaux 455, and Belgian Royal Decree of 29 May 2013). Studies performed in Italy were approved by the Institutional Ethical Board at the University of Rome Tor Vergata, according to the Guideline of the Italian Institute of Health (protocol n. 745/2022-PR). All the experimental procedures performed in Switzerland complied with the Swiss National Institutional Guidelines on Animal Experimentation and were approved by the Cantonal Veterinary Office Committee for Animal Experimentation. In all cases special attention was given to the implementation of the 3 R's, housing and environmental conditions and analgesia to improve the animals' welfare. Animal manipulation performed in Belgium was done according to the ECD protocol approved by the Institutional ethical committee of the KU Leuven and Cantonal Veterinary Office Committee for Animal Experimentation.

A 12 h light/dark cycle was used, and food and water were available ad libitum. The *Cyfip1*[+/-] mouse line, generated by gene trap at the Sanger Institute, UK, was kindly provided by Seth G.N. Grant. The gene trap cassette was inserted between exon 12 and 13. Molecular and behavioral characterization of the *Cyfip1*[+/-] mice was previously described in refs. 17,28. C57/Bl6 wild type (WT) and *Cyfip1*

heterozygous (*Cyfip1*$^{+/-}$) mice at embryonic and postnatal days (E15.5, P5, P15, P30) were used. *Cyfip1*$^{+/-}$ male were crossed to wild-type C57Bl6 females and both male and female mice were used for the analysis.

## In Utero Electroporation (IUE)

IUE was performed as described in La Fata et al. 2014[114]. In brief, pregnant mice at embryonic stage E15.5 were anesthetized by intramuscular injection with a solution containing 75 μg ketamine (Anesketin, Eurovet) and 1 μg of Medetomidin Hydrochloride (Dormitor, Pfizer) per g of body weight or with a mixture of oxygen ($O_2$) and Isoflurane (0.8 L/min of $O_2$ and 2 L/min of isoflurane). During the procedure, the uterine horns were exposed and a mixture of Fast Green (Sigma-Aldrich, #F7252) with the plasmid (pCAG-tdTomato, or a mix of pCAG-tdTomato and pCAG-EGFP-hCYFIP1 WT) (1 μg/μl) was microinjected into the lateral ventricle of E15.5 mouse embryos. Electric pulses (37 V, 5 pulses; 50 ms on, 950 ms off; ECM 630 electroporator, BTX, Harvard Apparatus) targeting the dorsal-medial part of the cortex were delivered. Antipamezol (Antisedan, Orion Pharma) injection (1.5 μg per g of body weight) was administered to wake up the animals that were anesthetized with ketamine and medetomidine. Progeny was analyzed at postnatal day 5, 15 and 30 (P5, P15 and P30). Mice were transcardially perfused with ice cold phosphate buffered saline and 4% paraformaldehyde and brains were post-fixed in 4% paraformaldehyde overnight at 4 °C.

The pCAG-tdTomato plasmid was kindly provided by Joris de Wit and has been previously described in DeNardo et al., 2012[115]. The pTagRFP-mito plasmid was purchased from Evrogen (#FP147). The pCAG-mito-GCaMP5G plasmid was purchased from Addgene (#105009). The pCAG-EGFP-hCYFIP1 WT was made by cloning the human CYFIP1 CDS (NM_014608) into the pCAG-EGFP (Addgene, #89684) plasmid using the XhoI and NotI restriction sites. Plasmid DNAs used in this study were produced using the Qiagen EndoFree Plasmid Maxi Kit (Qiagen, #12362).

## In vivo axonal growth and arborization analysis

Coronal sections from postnatal day 5 (P5) and 15 (P15) mice brains were obtained in a Leica vibratome VT1000S (100 μm thick) and slices incubated with DAPI (Invitrogen, #D1306) and mounted in Mowiol (Sigma-Aldrich, #S-81381) prior to visualization. To analyze axonal growth, z-stack tile images of P5 brain slices were acquired with a confocal microscope (Leica SP8; 10x objective, 1024 × 1024 pixels per image with 4 μm z-stack, using the LAS-X 3.0 software). Axonal length was quantified using Fiji (ImageJ) (version 1.49, NIH, USA). The "invasion index" of the axons growing in the contralateral cortex was calculated as the proportion of axons growing in each cortical bin, over those growing 300 μm from the corpus callosum (this normalization takes into account the transfection efficiency in each mouse).

To analyze axonal growth at P15, fluorescence images were taken with a florescence microscope (Olympus IX71; 4x objective). Axonal length was quantified using Fiji (ImageJ) (version 1.49, NIH, USA). Measurements in the contralateral hemisphere were normalized by the ipsilateral side, and length of the axons in the cortex by the total cortical thickness.

To analyze axonal arborization at P15 and P30 z-stack confocal images, from somatosensory areas with a low number of transfected axons, were acquired on a confocal microscope (20x objective, 1024 × 1024 pixel images with 1 μm z-stack). Images were acquired on a Nikon through a Hercules Type 1 AKUL/09/037 financing to W. Annaert or a Leica SP8 (Leica) confocal microscope using the LAS-X 3.0 software. Arborization was analyzed by manually tracing axons across the stack of optical images with Fiji (Image J) (version 1.49, NIH, USA). We counted the primary, secondary, tertiary, quaternary and total number of branches. Only branches that could be reliably assigned were counted.

## Acute organotypic brain slices preparation, TMRE loading and imaging

Organotypic slice from WT and *Cyfip1*$^{+/-}$ pups were prepared as follows: postnatal day 5 and 6 pups were sacrificed, and brains were placed in ice-cold artificial cerebrospinal fluid (ACSF) solution containing 125 mM NaCl (Sigma-Aldrich, #21115), 3 mM KCl (Sigma-Aldrich, #P9333), 1.25 mM $NaH_2PO_4$ (J.T. Baker, #1768), 10 mM Glucose (Sigma-Aldrich, #G6152), 26 mM $NaHCO_3$ (Sigma-Aldrich, # S6297), 1.5 mM $MgCl_2$ (Sigma-Aldrich, # M1028), and 1.6 mM $CaCl_2$ (Sigma-Aldrich, # 21115), continuously bubbled with 95% $O_2$ and 5% $CO_2$. Coronal brain slices (300 μm) were sectioned using a Leica VT1000 S vibratome and then placed in bubbled ACSF for 1 h at room temperature to allow recovery.

The slices were then incubated with 100 nM[116] TMRE (Enzo, #115532-52-0) in ACSF for 30 min at 34 °C, followed by washing in ACSF for 30 min at room temperature.

Imaging was performed using a Leica SP8 confocal microscope with a 25x immersion objective in a pre-warmed live imaging chamber and the LAS-X 3.0 software. The thickness of the optical section for the z-stack was 1 μm, and the total thickness of the acquired portion was 10 μm.

## Primary cortical neurons and microfluidic chambers

Mouse primary cortical neurons (E15.5) were prepared as previously described in Pasciuto et al., 2015[117]. In brief, mouse brains were removed, neocortices dissected and freed of meninges, treated with 0.025% trypsin (Gibco, #25200056), minced, and plated on poly-L-lysine (0,1 mg/ml, Sigma-Aldrich, #P2636) coated dishes, coverslips or Standard Neuron Devices microfluidic chambers (Xona Microfluidics, #SND450). Neurons were cultured in Neurobasal™ Medium (Gibco, #21103049) supplemented with 2% B-27™ Supplement (50X; Gibco, #17504044), 1mM L-Glutamine (200 mM; Thermo Fisher Scientific, #25030024) and 0.5% Penicillin-Streptomycin (10,000 U/mL; Gibco, #15140122). Standard Neuron Device microfluidic chambers (Xona Microfluidics, #SND450) were placed on poly-L-lysine coated coverslips and 200.000 neurons were plated in the soma compartment.

## Axonal growth in vitro

Axonal growth was analyzed by acquiring life bright field images of the axons growing in the axonal compartment at day in vitro (DIV) 3 and 4 (Olympus IX71; 10x objective, Leica DMi8). Growth rate of individual axons was measured by subtracting the length at DIV 4 and DIV 3 using Fiji (ImageJ) (version 2.16.0).

## Fluo-4 AM loading and imaging

The axoplasmic and cytosolic calcium concentration was measured using Fluo-4 AM (Thermo Fisher Scientific, #F14201). DIV 3 cultured neurons plated in Standard Neuron Device microfluidic chambers (Xona Microfluidics, #SND450) or in glass bottom microwell dishes (MatTek, #P35G-1.5-14-C) were incubated with 2 μM Fluo-4 AM (Thermo Fisher scientific, #F14201) in Neurobasal™ Medium without phenol red (Gibco, #12348017) supplemented with 2% B-27™ Supplement (50X; Gibco, #17504044), 1mM L-Glutamine (200 mM; Thermo Fisher scientific, #25030024) and 0.5% Penicillin-Streptomycin (10,000 U/mL; Gibco, #15140122) for 30 min at 37 °C. After incubation, neurons were imaged on a Leica AF6000 LX microscope (Leica) using the LAS-X 3.0 software.

Mitochondrial calcium concentration was measured by seeding cortical neurons on glass bottom microwell dishes (MatTek, #P35G-1.5-14-C) and transfecting them at DIV 1 with the pCAG-mitoGCaMP5G plasmid (Addgene, #105009). Neurons were transfected using Lipofectamine2000™ (Invitrogen, #11668027) following the manufacturer's protocol.

To obtain quantitative information on axoplasmic, cytosolic, and mitochondrial calcium concentrations ($[Ca^{2+}]$) the following $Ca^{2+}$ calibration formula was used: $[Ca^{2+}] = K_d*((F − F_{min})/(F_{max} − F))$, where F is the mean fluorescence intensity in a region of interest (ROI) (the axon for axoplasmic, or the total neuron for cytosolic analysis), $F_{min}$ is the fluorescence intensity recorded in the absence of calcium (achieved by substituting the Neurobasal™ medium (Gibco, #12348017) with HBSS (without calcium, magnesium, or phenol red; Gibco, #14175053) containing 6 mM EGTA), and $F_{max}$ is the fluorescence intensity observed in a ROI under calcium-saturated condition (obtained by treating cells with Ionomycin 1 µM (Sigma-Aldrich, #I9657) in Neurobasal™ medium without phenol red (Gibco, #12348017)). $K_d = 345$ nM is the dissociation constant for Fluo-4 AM, while $K_d = 460$ nM is the dissociation constant for mitoGCaMP5G[118]. Image analysis was performed using Fiji (ImageJ) (version 2.16.0).

## TMRE loading and imaging

The mitochondrial membrane potential was measured using TMRE (Enzo, #115532-52-0). DIV 3-4, DIV 7 and DIV 14 cultured neurons plated in Standard Neuron Device microfluidic chambers (Xona Microfluidics, #SND450) were incubated with 10 nM TMRE in Neurobasal™ Medium without phenol red (Gibco, #12348017) supplemented with 2% B-27™ Supplement (50X; Gibco, #17504044), 1 mM L-Glutamine (200 mM; Thermo Fisher scientific, #25030024) and 0.5% Penicillin-Streptomycin (10,000 U/mL; Gibco, #15140122) for 30 min at 37 °C. After incubation, neurons were imaged on a Leica Stellaris 8 and Leica SP8 confocal microscopes, equipped with a K5 sCMOS camera and a HC PL APO 63x/1.40 oil CS2 objective (Leica), and using the LAS-X 3.0 software.

## ATP measurement

ATP levels were measured using an ATP assay kit (Abcam, #ab83355) in accordance with the manufacturer's instructions. In brief, DIV 3, DIV 7 and DIV 14 neurons, and P5, P15 and P30 cortex were lysed in the supplied buffer, and proteins were removed using the Deproteinizing Sample Preparation Kit – TCA (Abcam, #ab204708). Fluorescence was recorded at excitation/emission wavelengths of 535/587 nm to quantify ATP levels, using Varioskan™ LUX Multimode Microplate Reader (Thermo Fisher Scientific) equipped with SkanIt software (version 7.0.1).

## Neuronal pTagRFP-mito transfection, live imaging, and mitochondrial motility analysis

DIV 0 cortical neurons were electroporated in suspension with the pTagRFP-mito plasmid (Evrogen, #FP147) in a solution of 10% PBS 10X, 5.5 mM Glucose and 1 mM $CaCl_2$, using an ECM 830 Square Wave Electroporation System (BTX, Harvard Apparatus). $2.5 × 10^6$ cells were resuspended in 25 µl of electroporation buffer plus 2 µg of pTagRFP-mito plasmid (Evrogen, #FP147), inserted in a 1 mm gap cuvette (BTX; #45-0134) and electroporated with a 112 V pulse of 5 msec duration. After resuspending the cells in 500 µl of prewarmed media, the neurons were seeded on glass bottom microwell dishes (MatTek, #P35G-1.5-14-C) and imaged 3-, 7- and 14-days post-electroporation. We identified the axon as the longest neurite based on previous studies describing the establishment of neuronal polarity and axonal specification in vitro[119]. Time-lapse images were taken at 2 s intervals for a total of 400 s, using a Leica AF6000 LX microscope (Leica) equipped with a Leica DFC350 FX camera and a Leica HCX PL FLUOTAR 100x/1.30-0.60 Oil objective (Leica), using the LAS-X 3.0 software. Kymographs were generated from the time-lapse movies and analyzed with Kymolyzer[120] in Fiji (ImageJ) (version 2.16.0). The percentage of motile mitochondria was calculated as the ratio between the number of motile mitochondria over the total number of mitochondria along each axon analyzed.

## Immunofluorescence and mitochondrial density analysis

Primary cortical neurons were fixed with 4% PFA/SEM (PFA (Electron Microscopy Sciences, #15710), 60 mM Sucrose (Merk, #107687), 3 mM EGTA (Sigma-Aldrich, #E4378) and 0.1 mM $MgCl_2$ (Sigma-Aldrich, #M1028) for 10 min at RT. Cells were permeabilized and blocked with 0.3% Triton X-100 (Sigma-Aldrich; #93443), 0.5% BSA (Sigma-Aldrich; #A6003) and 2% Goat Normal Serum (Sigma-Aldrich; #G9023) for 1 h at RT prior to antibody incubation. The following primary antibodies were used: mouse anti-βIII Tubulin (1:200, BioLegend, #801201), rabbit anti-Ca$_V$1.2 (CACNA1C) (1:100, Alomone Labs, #ACC003), rabbit anti-Ca$_V$2.3 (CACNA1E) (1:100, Alomone Labs, #ACC006), and rabbit anti-Ca$_V$3.3 (CACNA1I) (1:100, Alomone Labs, #ACC009). Neurons were incubated with primary antibodies overnight at 4 °C and Alexa 488 (Life Technologies, #A11029) and Alexa 555 (Life Technologies, #A21428) conjugated secondary antibodies for 1 h at RT. DAPI (Invitrogen, #D1306) was used for nuclei visualization.

For the analysis of mitochondrial density, DIV 0 cortical neurons were electroporated in suspension with the pTagRFP-mito plasmid (Evrogen, #FP147) and plated on coverslips. At DIV 3, neurons were fixed with 4% PFA/SEM for 10 min at RT and then stained with a mouse anti-βIIITubulin (1:200, BioLegend, #801201) and DAPI (Invitrogen, #D1306) for nuclei visualization. All images were acquired using a Leica AF6000 LX microscope (Leica) equipped with a Leica DFC350 FX camera and a Leica HCX PL FLUOTAR 100x/1.30-0.60 Oil objective (Leica), using the LAS-X software.

The analysis was done using Fiji (ImageJ) (version 2.16.0). by drawing concentric circles around the soma with 10 µm distance between them. We identified the axon as the longest neurite and counted the number of mitochondria present in each segment. The total mitochondria number was normalized by the full length of the axon. Mitochondrial number was also counted along dendrites and normalized by the total dendritic length.

## Electron microscopy

DIV 0 cortical neurons were plated in Standard Neuron Device microfluidic chambers (Xona Microfluidics, #SND450) as described above. At DIV 3, axons adhered to the slides were fixed with 2.5% glutaraldehyde (EMS, #16220) plus 2% paraformaldehyde (Sigma-Aldrich, #P6148) in 0.1 M sodium cacodylate (Electron Microscopy Sciences, #12300) buffer pH 7.4 overnight at 4 °C. Subsequently the samples were postfixed with 1% osmium tetroxide plus potassium ferrocyanide 1% in 0.1 M sodium cacodylate buffer for 1 h at 4 °C. After three water washes, samples were dehydrated in a graded ethanol series and embedded in an epoxy resin (Sigma-Aldrich #46345). Ultrathin sections (60−70 nm) were obtained with a Leica Ultracut EM UC7 ultramicrotome, counterstained with uranyl acetate and lead citrate and viewed with a Tecnai G2 (FEI) transmission electron microscope operating at 100 kV. Images were captured with a Veleta (Olympus Soft Imaging System) digital camera.

Mitochondrial shape descriptors, such as area, major and minor axis, and roundness were calculated using Fiji (ImageJ) (version 2.16.0).

## RNA immunoprecipitation, RNA isolation and RT-qPCR

RNA immunoprecipitation was performed on extracts from cortical neurons prepared as described previously[21]. Protein extracts were incubated with protein G Dynabeads (Invitrogen, #10004D) coated with anti-CYFIP1 antibody (Sigma-Aldrich, #AB6046), with anti-HuD/Elavl4 (Santa Cruz, #sc-48421) or with anti-HuR/Elavl1 (Santa Cruz, #sc-5261) for 1 h at 4 °C.

Bound RNA was eluted with TRIzol™ reagent (Invitrogen, #15596018) and isolated according to the manufacturer's description.

For whole cell analysis, RNA was isolated from WT and *Cyfip1*$^{+/-}$ DIV 3 cortical neurons using TRIzol™ reagent (Invitrogen, #15596018).

cDNA was synthesized using M-MLV reverse transcriptase (Invitrogen, #28025-013) and random primers (Promega, #C1181) and quantified by quantitative real time-PCR (RT-qPCR), performed on StepOnePlus™ Real-Time PCR System (Thermo Fisher Scientific) using the Sso Advanced Universal SYBR Green Supermix (Bio-Rad, #1725271). mRNA levels of mouse *Microtubule Associated Protein 1B* (*Map1b*), *Discs large MAGUK scaffold protein 4* (*Dlg4*), *Calcium Voltage-Gated Channel Subunit Alpha1 C* (*Cacna1c*), *Calcium Voltage-Gated Channel Subunit Alpha1 E* (*Cacna1e*), *Calcium Voltage-Gated Channel Subunit Alpha1 I* (*Cacna1i*), *Calcium Voltage-Gated Channel Auxiliary Subunit Beta 3* (*Cacnb3*) and *Calcium Voltage-Gated Channel Auxiliary Subunit Gamma 2* (*Cacng2*) were calculated in relative abundances compared to *Hypoxanthine Phosphoribosyltransferase* (*Hprt1*) or *Histon 3* (*H3f3*) mRNA level, and expressed as a fold change over WT levels. Primer sequences are described in Table S1.

For HuD/Elavl4 and HuR/Elavl1 RNA-IP, the enrichment of the calcium channels mRNA is expressed as a ratio over the positive control *mDlg4* enrichment rate, previously identified in the Hu protein complex and used as a positive control.

The quantitative PCR for mtDNA content was adapted from Kanellopoulos et al. 2020[40]. The whole DNA (genomic and mitochondrial) was purified from neurons, and the extracted DNA was used for quantitative PCR, mixed with primers and Sso Advanced Universal SYBR Green Supermix (Bio-Rad, #1725271). The primer pairs used to amplify a genomic DNA fragment, corresponding to rpS23 and a mitochondrial DNA fragment, corresponding to mt16S, are described in Table S1. The mtDNA content was calculated as mt16S/rpS23 ratio.

## Membrane-enriched fraction preparation and Western Blotting

DIV 3 cortical neurons were lysed in RIPA buffer containing 150 mM NaCl (Merck, # 106404), 50 mM Tris-HCl (Millipore, #108382) pH 7.4, 1% Triton X-100 (Sigma-Aldrich, #93443), 1% sodium deoxycholate (Sigma-Aldrich, #D6750), 1 mM EDTA (J.T. Baker, #1073), Protease Inhibitor Cocktail (PIC; Sigma-Aldrich, #P8340), and Phosphatase Inhibitor (PhosSTOP; Roche, #4906845001). The lysate was kept on ice for 10 min and then centrifuged at 12000 *g* for 10 min. The recovered supernatant was used for Western Blot analysis.

For the membrane-enriched fraction preparation DIV 3, DIV 7 and DIV 14 cortical neurons were lysed with a buffer containing 0.32 M Sucrose (Merk, #107687), 10 mM HEPES (Gibco, #15630-056) and 400 μM Phenylmethanesulfonyl Fluoride (PMSF; Sigma-Aldrich, #78830). The protein extract was centrifuged at 1000 *g*. The recovered supernatant was further centrifuged at 13800 *g* to obtain a pellet containing the membrane-enriched fraction. The pellet was lysed in RIPA buffer. The neuronal protein extracts and the membrane-enriched fractions were separated by SDS-PAGE electrophoresis and blotted on a PVDF membrane (Merck, #GE10600023). EveryBlot blocking buffer (BioRad, #12010947) was used prior to antibody incubation. Membranes were incubated with the following antibodies rabbit anti-CYFIP1 (1:1000; Sigma-Aldrich, #AB6046), rabbit anti-Ca$_V$1.2 (CACNA1C) (1:500, Alomone Labs, #ACC003), rabbit anti-Ca$_V$2.3 (CACNA1E) (1:500, Alomone Labs, #ACC006), rabbit anti-Ca$_V$3.3 (CACNA1I) (1:500, Alomone Labs, #ACC009), rabbit anti-Stargazin (Ca$_V$γ2/CACNG2) (1:500, Alomone Labs, #ACC012) and rabbit anti-Ca$_V$β3 (CACNB3) (1:500, Alomone Labs, #ACC008), HuD/Elavl4 (1:500, Santa Cruz, #sc-48421) and HuR/Elavl1 (1:500, Santa Cruz, #sc-5261). A secondary antibody anti-rabbit HRP (1:2500, Cell Signaling Technology, #7074S) or anti-mouse HRP (1:2500, Cell Signaling Technology, #7076S) was used. Proteins were revealed using Clarity Max Western ECL Substrate (Bio-Rad, #1705062) and the imaging system LAS-4000 mini (GE Healthcare). Quantification was performed using the ImageQuant software (GE Healthcare). Coomassie staining was used as normalizer, and the protein levels were expressed as a fold change over WT.

## Prediction of mRNA binding sites

The frequency of HuD and HuR binding sites on calcium channel subunits mRNAs has been predicted using RBPmap[67,121]. Graphs showing the predicted frequency of binding sites every 200nt were generated using R 4.1.

## Actinomycin D assay

The assay was performed using WT and *Cyfip1*[+/-] DIV 3 cortical neurons. A stock solution of 5 mg/ml of Actinomycin D (Gibco, #11805-017) was prepared in DMSO (Sigma Aldrich, #D2650). Cells were treated with Actinomycin D (Gibco, #11805-017) at a final concentration of 3 μg/ml for 0, 2, and 4 h. Control cells received an equivalent volume of DMSO (Sigma Aldrich, #D2650). At the end of the treatment cells were lysed with TRIzol™ reagent (Invitrogen, #15596018) and the total RNA was isolated according to the manufacturer's description. RT-qPCR was performed as described above.

## Statistics

Statistical analyses were performed with GraphPad Prism 9. The statistical tests used are listed in the respective Figure legends. For all analysis, *p*-values < 0.05 were considered significant and annotated as follows: *$p < 0.05$, **$p < 0.01$, ***$p < 0.001$, ****$p < 0.0001$.

## Reporting summary

Further information on research design is available in the Nature Portfolio Reporting Summary linked to this article.

## Data availability

Source data are provided with this paper.

## Code availability

This paper does not report original code.

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

## Acknowledgements

We thank Joanna Viguie and Melanie Reinero for technical assistance and Diane Estade and Manuela Novelli for administrative support. The initial part of this work started at the KU Leuven and we are thankful to Karin Jonckers and Jonathan Royaert for technical assistance. We are grateful to Annette Gartner, Esperanza Fernández and all members of the Bagni lab for helping during the development of this project, contributing to stimulating discussions and for sharing preliminary data. We are grateful to Jean-Yves Chatton for his insights into calcium homeostasis and excellent discussions. This work was supported by SNSF 310030 - 182651 (Switzerland), ERANET-NEURON Joint Transnational Research Projects on Sensory Disorders 2020-088 (Switzerland), Etat de Vaud (VD, Switzerland), PRIN-MUR N. 20227JA8R3 (Italy), Telethon GGP20137 (Italy), Fondazione Italiana per l'Autismo (Italy) and MNESYS PE0000006, DN. 1553 11.10.2022 MUR (Italy). N.D.I. was recipient of an FWO aspirant PhD fellowship 11Y2416N (Belgium). Figures were created with BioRender.com. During the preparation of this manuscript the authors used ChatGPT to improve language and readability. After using this tool, the authors reviewed the content and take full responsibility for the content of the publication.

## Author contributions

Conceptualization, N.D.I., C.R., T.A. and C.B.; Experiments performed by C.R., N.D.I., M.M. and F.C.; T.A., helped with data analysis and manuscript preparation.

## Competing interests

The authors declare no competing interests.
