## [Transparent Peer Review file · Nature Communications]

CYFIP1 governs the development of cortical axons by modulating calcium availability

Corresponding Author: Professor Claudia Bagni

Version 0:

Reviewer comments:

Reviewer #1

(Remarks to the Author)

Summary

In this manuscript, the authors investigate the significance of cytoplasmic FMR1-interacting protein (Cyfip1) in cortical development. They find a role for Cyfip1 in regulation of mRNA stability of selected transcripts (possibly via its known binding partners HuD and HuR), leading to the loss of subsequent protein expression. They demonstrate that esp. the loss of voltage-sensitive calcium channel subunit expression modulates intracellular calcium levels. This change in calcium availability influences the mitochondrial motility and function, which is suggested to be contribute to cortical axonal development and circuit formation. The authors undertake a straightforward and simple approach for investigating their research questions and report their findings in a coherent manner. However, there is a tendency of the authors drawing strong conclusions from their experiments which do not provide such a wider scope. For the final hypothesis to be more convincing, the authors might need to fill more gaps in their flow of experiments. Overall, the manuscript can benefit from the following suggestions.

Major points:

1. While the authors use a cDNA rescue construct in the beginning, this is not consistently done throughout the manuscript. This is a missed opportunity, as mutation of CYFIP1 that lead to a reduced interaction with the FMRP complex (as in their previous publication PMID 18805096) could serve as the ideal control to shown that their effect of CYFIP1 loss is indeed via interaction with this complex.
2. Along similar lines, no experiments directly address whether HuD/HuR are involved in the phenotype of CYFIP^{+/-} neurons, it is only suggested due to the presence of HuD/HuR motifs in the RNAs bound to CYFIP1.
3. Same for the specific voltage-sensitive calcium channel subunits affected, could overexpression of them also rescue the phenotype?
4. Not only mitochondrial movement, but also mitochondrial shape is sensitive to cellular Ca²⁺ concentration (PMID: 18838687, 29694881). The authors use the number of mitochondria as a measure for their abundance, but the actual mitochondrial mass could be similar if the increased mitochondrial density in CYFIP^{-/-} neurons was made up of smaller mitochondria. The authors should quantify the length of mitochondria and factor this into their conclusions.
5. Next to the reduction in axon length, there appear to be less labelled axon in Fig 1b. Is this a function of the slower growth or is there also some form of neuronal cell death triggered?
6. The direction of mitochondrial motility (anterograde vs. retrograde) is thought to be indicative of their health. Considering the lower membrane potential of Cyfip1^{+/-} axonal mitochondria, it would be interesting to measure the directionality of mitochondrial movement.
7. Mitochondrial Ca²⁺ uptake also depends on the mitochondrial membrane potential. How does the mitochondrial Ca²⁺ content alter upon loss of CYFIP? Could increased mitochondrial Ca²⁺ uptake underlie the reduction in cytosolic Ca²⁺ and the reduced membrane potential?
8. How do the authors explain the unaltered mitochondrial motility in WT neurons when compared between ionomycin and DMSO? If ionomycin treatment does not cause a decrease in WT mitochondria motility, then how are the authors convinced that the reduction in Cyfip1^{+/-} mitochondria motility is indeed due to the calcium availability provided by ionomycin? The authors should consider validating the effect of ionomycin in actually causing an increase in intracellular calcium levels.

Minor point:

Normalization of qPCR/Western Blot data to the respective WT control should be stated in the methods/figure legends. E.g.

for the RT-PCR it is only stated to be normalized to mH3f3, but clearly is also normalized to the WT control.

Reviewer #2

(Remarks to the Author)

This study aims to investigate the impact of *Cyfp1* haploinsufficiency, a gene related to neurodevelopmental disorders (NDD), on neuronal development. The research builds on the authors' previous study published in *Nat Commun* in 2019, which demonstrated that *Cyfp1* haploinsufficiency caused changes in both anatomical and functional connectivity in cortical circuits of mice. The new manuscript delves deeper into the underlying molecular mechanisms, with a focus on axonal development. Importantly, the manuscript suggests that *Cyfp1* plays a crucial role in early axonal development, and insufficient calcium uptake is the critical mechanism that governs this process. *CYFIP1* regulates mRNA stability for specific voltage-gated calcium channel subunits, which explains the reduced calcium concentration in *Cyfp1* mutant cells. However, in order to properly support the manuscript's current conclusions, the data in several places (detailed below) would need substantial strengthening, likely involving new experiments.

Major points:

1. In vivo evidence:

The paper focuses on the evidence that *Cyfp1* haploinsufficiency leads to a reduction in intracellular calcium, which in turn affects mitochondria motility, density, membrane potential, and axonal growth. However, all experiments were conducted in vitro. It is important to note that both calcium homeostasis and mitochondrial function are highly sensitive to culture conditions, which raises questions about the relevance of these findings in vivo. Therefore, demonstrating with functional experiments the in vivo relevance of calcium level and mitochondria features caused by *Cyfp1* haploinsufficiency would greatly strengthen the key conclusions of this paper.

2. Cell identity in vitro & cell type-specific pathology:

The authors observed that in *Cyfp1* mutant mice, growth and arborization of axons in the superficial layer neurons were slower compared to control mice (Fig 1 & S1). For later in vitro experiments, the authors used E15.5 embryonic brains, which potentially include both deep layer and superficial layer pyramidal neurons with different birth dates. Therefore, it is unclear which cell types were analyzed in these experiments. Also, it is not clear whether there is any difference in phenotype between deep-layer and superficial-layer neurons due to *Cyfp1* haploinsufficiency.

3. Mitochondria function in *Cyfp1* mutant

The authors found reduced mitochondria membrane potential in *Cyfp1* mutant through TMRM imaging and concluded that "Our results suggest that *CYFIP1* regulates mitochondria functions (lines 179-181)". It would be valuable to conduct more in-depth research on mitochondria properties that are related to axonal development (ref 43) in order to strengthen their conclusion. This would also help to understand the difference of this study and a previous one where *Cyfp1* haploinsufficient flies displayed higher mitochondria membrane potential and oxygen consumption rate (ref 40).

4. Developmental changes

The authors conducted experiments on all in vitro parts and identified phenotypes at DIV 3 and 4. However, further investigation is needed to determine whether these phenotypes will continue to persist at a later stage or if they will disappear, similar to the axonal morphology phenotype in vivo. This information could be extremely valuable in advancing our understanding of the pathologies caused by *Cyfp1* haploinsufficiency, as well as the relationships among the variable endophenotypes caused by this condition.

Minor points:

5. It would be helpful to annotate which part is axon or dendrites in Figure 2B and 3C.

6. To strengthen the author's argument regarding the focus on axonal mitochondria, it would be beneficial to include additional results that use more rigorous criteria for identifying axons, such as AIS staining, instead of the criteria described by the author in lines 752-754.

7. It would be helpful to provide evidence that treatments with chemical compounds such as IONO or BAPTA-AM alter cytoplasmic calcium concentration in axons.

8. Each step of neuronal development and maturation influences the other. It would be valuable to investigate other neuronal developmental parameters, such as dendrite growth, in *Cyfp1* mutants.

Reviewer #3

(Remarks to the Author)

In this manuscript, the authors examine the role of *Cyfp1*, a protein associated with higher risk of ASD and SCZ, in the regulation of axonal growth in cortical neurons using *cyfp1*^{+/-} mice as a model. This is an important question as defects in axonal growth and branching are emerging as common defects in multiple genetic models of ASD, leading to dysfunctional connectivity and behavioral phenotypes. *Cyfp1* function has been extensively studied in dendrites and spine formation/plasticity in neurons, but its role in axons is less clear.

Here the authors demonstrated that CPN neurons display a delay in axonal development in vivo and in vitro, an effect that appears cell autonomous. This defect correlates with an increase in mitochondrial density and motility, as well as a decrease in mitochondrial membrane potential, all signs that are coherent with a delay in axonal maturation. Interestingly, the authors show a decreased concentration in intracellular calcium when CYFIP1 levels are reduced. Mechanistically, they provided evidence of a reduced expression of specific calcium channel subunits due to a decreased stability of their respective mRNAs in *cyfip1*^{+/-} neurons. Finally, they reported that increasing intracellular calcium concentration using ionomycin was sufficient to rescue the defects in axonal growth, mitochondrial motility and mitochondrial membrane potential observed in *cyfip1*^{+/-} neurons.

The work is well written, and the images are clear. The results nicely demonstrate a new role for *Cyfip1* in the regulation of intracellular calcium concentration during axonal growth. A few additional experiments should be done to better understand how the phenotypes observed are correlated.

Main points:

- It remains unclear to me how an overall increase in mitochondrial density observed in axons from *cyfip1*^{+/-} neurons is not concomitant with an increase in the total mitochondrial mass. Do the authors believe that this effect is due to the same number of mitochondria present in a smaller volume? This conclusion could be confirmed by performing additional experiments and quantify the expression of mitochondrial resident/structural proteins in *cyfip1*^{+/-} neurons compared to control? Maybe by Western blot experiments and normalizing on the cytoskeletal?
- Linked to the previous comment, is mitochondrial morphology affected in *cyfip1*^{+/-} neurons? In some pictures, mitochondria appear smaller in *cyfip1*^{+/-} neurons.
- Or does it mean that mitochondria are not well distributed in *cyfip1*^{+/-} neurons? If so, would it possible that *cyfip1*^{+/-} neurons display a reduction in mitochondrial density in the somatodendritic compartment? Or maybe mitochondria are not well distributed in axons? The authors should analyze mitochondria density and morphology in axons using microfluidic device, allowing a better resolution of the more distal portion.
- If additional features emerge from this analysis, the authors could test if ionomycin rescues the phenotype in *cyfip1*^{+/-} neurons.
- The defect in mitochondrial motility in *cyfip1*^{+/-} neurons should be better characterized with additional information regarding the directionality (anterograde versus retrograde), the % of oscillatory and % of directed transport.
- The authors suggest that CYFIP1 could control the stability of specific mRNAs through its binding to HuD and/or HuR proteins. The authors should analyze the expression of these two RBPs in *cyfip1*^{+/-} neurons and determine if their expression is changed when CYFIP1 is reduced.

Version 1:

Reviewer comments:

Reviewer #1

(Remarks to the Author)

The authors have responded to all comments adequately. This is a beautiful study and I recommend accepting the manuscript.

Reviewer #2

(Remarks to the Author)

The authors have addressed all of our questions, and the paper has been significantly improved. Therefore, it is now suitable for publication.

Reviewer #3

(Remarks to the Author)

I have carefully read the authors' responses and the revised manuscript. I am satisfied that the authors have adequately addressed all of my main concerns. The revisions have improved the clarity and quality of the work.

I therefore recommend that the manuscript be accepted for publication.

Reviewer #4

(Remarks to the Author)

I co-reviewed this manuscript with one of the reviewers who provided the listed reports. This is part of the Nature Communications initiative to facilitate training in peer review and to provide appropriate recognition for Early Career

Researchers who co-review manuscripts.

Ricci et al.,

RESPONSE TO REFEREES

Reviewer #1 (Remarks to the Author):

Summary

In this manuscript, the authors investigate the significance of cytoplasmic FMR1-interacting protein (Cyfip1) in cortical development. They find a role for Cyfip1 in regulation of mRNA stability of selected transcripts (possibly via its known binding partners HuD and HuR), leading to the loss of subsequent protein expression. They demonstrate that esp. the loss of voltage-sensitive calcium channel subunit expression modulates intracellular calcium levels. This change in calcium availability influences the mitochondrial motility and function, which is suggested to be contribute to cortical axonal development and circuit formation. The authors undertake a straightforward and simple approach for investigating their research questions and report their findings in a coherent manner. However, there is a tendency of the authors drawing strong conclusions from their experiments which do not provide such a wider scope. For the final hypothesis to be more convincing, the authors might need to fill more gaps in their flow of experiments. Overall, the manuscript can benefit from the following suggestions.

Major points:

1. While the authors use a cDNA rescue construct in the beginning, this is not consistently done throughout the manuscript. This is a missed opportunity, as mutation of CYFIP1 that lead to a reduced interaction with the FMRP complex (as in their previous publication PMID 18805096) could serve as the ideal control to shown that their effect of CYFIP1 loss is indeed via interaction with this complex.

We thank the reviewer for this important point. As the reviewer points out, in a previous publication we were able to disentangle the contribution of each CYFIP1 function (actin remodeling or translational control) in regulating spine morphology. However, CYFIP1, in addition to FMRP also interacts with other RNA binding proteins such as HuD, HuR, Caprin etc (De Rubeis et al., 2013) some of those involved in the stability of the mRNAs. In this study, our data suggests that CYFIP1 regulation of intracellular neuronal calcium is independent of those two canonical functions, one of which includes FMRP. Our data indicates that CYFIP1 regulates mRNA stability of calcium channels, through its binding with HuD/Elavl4 and HuR/Elavl1 (Figures 4 and 5), a mechanism that would be independent of FMRP. To further confirm this, we have: 1) verified CYFIP1 binding to HuD and HuR by co-immunoprecipitation (Figure S4, new panel f), and 2) conducted RNA immunoprecipitation of HuD and HuR in WT and *Cyfip1*^{+/-} cortical neurons to assess their binding of *mCacna1c*, *mCacna1e* and *mCacna1i*. Our results demonstrate that both, HuD and HuR, bind *mCacna1c*, *mCacna1e*, and *mCacna1i*. Importantly, we observed reduced binding of HuD and

HuR to *mCacna1c* in *Cytip1*^{+/-} neurons compared with WT (Figure 5, new panels d and e). This finding supports our hypothesis that HuD/HuR binding to these calcium channel mRNAs may play a role in regulating their stability, which appears to be diminished under *Cytip1*^{+/-} deficient conditions. Altogether, our data demonstrate that CYFIP1 regulates the stability of calcium channel mRNAs through its interaction with HuD and HuR, rather than controlling their translation via FMRP.

We have included the new panels below for easier reference.

Figure S4 panel f: Representative Western blot of CYFIP1 protein immunoprecipitation from DIV 3 wild-type cortical neurons, showing the detection of HuD/Elavl4 and HuR/Elavl1 proteins.

Figure 5 panels d and e: **d** HuD/Elavl4 RNA immunoprecipitation (RNA-IP) from DIV 3 WT and *Cytip1*^{+/-} cortical neurons. Histograms represent calcium channel mRNAs enrichment, calculated as the ratio over the positive control (*mDlg4*) (WT n=4, *Cytip1*^{+/-} n=4; Multiple Mann-Whitney test; *mCacna1c* mRNA p=0.0285, *mCacna1e* mRNA p=0.0285, *mCacna1i* mRNA p=0.8857). **e** HuR/Elavl1 RNA-IP from DIV 3 WT and *Cytip1*^{+/-} cortical neurons. Histograms represent calcium channel mRNAs enrichment, calculated as the ratio over the positive control (*mDlg4*) (Multiple Mann-Whitney test; *mCacna1c* mRNA p=0.0310, *mCacna1e* mRNA p=0.2270, *mCacna1i* mRNA p=0.6289).

2. Along similar lines, no experiments directly address whether HuD/HuR are involved in the phenotype of CYFIP1^{+/-} neurons, it is only suggested due to the presence of HuD/HuR motifs in the RNAs bound to CYFIP1.

Following the reviewer's suggestion, we have conducted additional experiments demonstrating the following: 1) CYFIP1 binds HuD and HuR (Figure S4, new panel f), and 2) HuD and HuR bind to *mCacna1c*, *mCacna1e* and *mCacna1i* mRNAs, with reduced binding of HuD and HuR to *mCacna1c* and *mCacna1e* in *Cytip1^{+/-}* neurons (Figure 5, panels d and e). These findings further support the notion that CYFIP1 regulates the stability of these calcium channels through its interaction with HuD and HuR.

3. Same for the specific voltage-sensitive calcium channel subunits affected, could overexpression of them also rescue the phenotype?

Following the reviewer's suggestion, to investigate the role of voltage gated calcium channels (VGCCs) in the delayed axonal growth phenotype of *Cytip1^{+/-}* cortical neurons, we treated DIV 3 cortical neurons with the VGCCs agonists Bay-K-8644 and Nefiracetam and monitored their growth over 24h. Bay-K-8644 specifically activates L-type calcium channels, whereas Nefiracetam has a broader affect, targeting both L- and N-type calcium channels. CACNA1C, the most affected channel in *Cytip1^{+/-}* cortical neurons, is an L-type calcium channel, and is therefore activated by both Bay-K-8644 and Nefiracetam. Our results demonstrate that treatment with either agonist effectively rescues the delayed axonal growth phenotype observed in *Cytip1^{+/-}* neurons (Figure 6 new panel d).

We have included the new panel below for easier reference.

Figure 6 panel d: Left, representative images of axons growing in the axonal compartment of microfluidic devices at DIV 3 and DIV 4 for WT and *Cytip1^{+/-}* neurons treated with DMSO, (S)-(-)-Bay-K-8644 (50 μM) and Nefiracetam (10 μM). Scale bar 200 μm. Growth rate of each individual axon was quantified as represented in the scheme (black line represents DIV 3 and red line DIV 4). Right, average axonal growth rate in WT and *Cytip1^{+/-}* neurons treated with DMSO, (S)-(-)-

Bay-K-8644 and Nefiracetam (WT n=5 embryos, 142 DMSO-treated axons, 190 Bay-K-8644-treated axons and 189 Nefiracetam-treated axons, and *Cyfp1*^{+/-} n=7 embryos, 150 DMSO-treated axons, 258 Bay-K-8644-treated axons and 207 Nefiracetam-treated axons; mean ± SEM; Two-Way ANOVA, $F_{(2,30)} = 5.8$, $p=0.0070$; genotype effect $p=0.0003$, treatment effect $p<0.0001$, interaction $p=0.0070$).

Additionally, it is important to highlight that VGCCs play a crucial role in generating spontaneous regenerative calcium transients (SRCaTs) in developing cortical neurons, with these transients being particularly pronounced in axons (Kamijo et al 2018). Knockout of endogenous Cav1.2 (*mCacna1c*) in primary cortical neurons has been shown to reduce neurite length, indicating that Cav1.2-mediated calcium signaling promotes neurite elongation (Kamijo et al 2018), and further supporting our hypothesis. This information was already mentioned in the discussion section (lines 449-451).

4. Not only mitochondrial movement, but also mitochondrial shape is sensitive to cellular Ca²⁺ concentration (PMID: 18838687, 29694881). The authors use the number of mitochondria as a measure for their abundance, but the actual mitochondrial mass could be similar if the increased mitochondrial density in CYFIP^{-/-} neurons was made up of smaller mitochondria. The authors should quantify the length of mitochondria and factor this into their conclusions.

We thank the reviewer for rising this crucial point. In response to the reviewer's suggestion, we conducted electron microscopy on DIV 3 cortical neurons *in vitro* and assessed the length and shape of axonal mitochondria. Our findings indicate that both the area and the major axis of mitochondria are increased in *Cyfp1*^{+/-} axons. However, the roundness is not significantly different. Altogether, these results suggest that mitochondria in *Cyfp1*^{+/-} axons are slightly larger and more elongated than in WT. This data has been included in Figure 3 and incorporated into the results and discussion of the manuscript.

Figure 3 panel e: Left, representative electron microscopy images of mitochondria along WT and *Cyfp1*^{+/-} DIV 3 cortical axons. Scale bar 250 nm. Right, histograms showing average mitochondrial area, major axis, minor axis and roundness, expressed as percentage over WT.

(WT n=12, *Cyfp1*^{+/-} n=13; mean ± SEM; Mann–Whitney test; mitochondria area p=0.0055, major axis p=0.0398, minor axis p=0.2101, roundness p=0.1519).

5. Next to the reduction in axon length, there appear to be less labelled axon in Fig 1b. Is this a function of the slower growth or is there also some form of neuronal cell death triggered?

The number of labeled axons depends on the number of neurons that were electroporated. During *in utero* electroporation, only a fraction of neuronal progenitors located at the ventricular zone will take up the plasmid. Only neurons derived from those progenitors will contain the plasmid, express tdTomato, and thus be visible for analysis. To ensure our measurements are independent of the number of electroporated neurons or labeled axons, we: in panel a, normalize the number of axons in each bin by the total number of axons in bin 1 (as described in methods). In panel b, axonal growth was measured as a whole, rather than by individual axons, and was normalized to the total length of the cortex.

6. The direction of mitochondrial motility (anterograde vs. retrograde) is thought to be indicative of their health. Considering the lower membrane potential of *Cyfp1*^{+/-} axonal mitochondria, it would be interesting to measure the directionality of mitochondrial movement.

Following the reviewer's suggestion, we measured both mitochondrial length and speed of movement in anterograde and retrograde directions. None of these parameters showed significant differences between *Cyfp1*^{+/-} and WT. We have included this data in Figure S2 panel b.

Figure S2 panel b: Representative histograms showing forward length, forward speed, backward length, backward speed and reverse direction frequency of mitochondria movement along axons of WT and *Cyfp1*^{+/-} DIV 3 cortical neurons (WT n=11 embryos, 53 axons, *Cyfp1*^{+/-} n=8 embryos, 31 axons; mean ± SEM; Mann-Whitney test, forward length p=0.3511, forward speed p=0.9678, backward length p=0.2723, backward speed p=0.6574, reverse direction frequency p=0.2375).

7. Mitochondrial Ca²⁺ uptake also depends on the mitochondrial membrane potential. How does the mitochondrial Ca²⁺ content alter upon loss of CYFIP? Could increased mitochondrial Ca²⁺ uptake underlie the reduction in cytosolic Ca²⁺ and the reduced membrane potential?

To assess mitochondrial Ca²⁺ content, we transfected WT and *Cyfp1*^{+/-} neurons with the pCAG-mitoGCaMP5G plasmid. The analysis of mitochondrial Ca²⁺ levels in axons showed a significant reduction in DIV 3 *Cyfp1*^{+/-} neurons compared to WT (Figure 3 panel b). These findings suggest that diminished mitochondrial Ca²⁺ uptake and membrane potential are most likely a consequence of reduced intracellular calcium levels rather than a primary driving factor.

Figure 3 panel b: Left, representative images of mitoGCaMP5G fluorescence intensity in the axonal mitochondria of WT and *Cyfp1*^{+/-} DIV 3 and DIV 4 cortical neurons. Scale bar 20 μ m. Inset scale bar 5 μ m. Right, histogram showing the axonal mitochondrial calcium concentration [Ca²⁺] expressed as a percentage over WT (WT n=8 embryos, *Cyfp1*^{+/-} n=8 embryos; mean \pm SEM; Mann-Whitney test, p=0.0030).

8. How do the authors explain the unaltered mitochondrial motility in WT neurons when compared between ionomycin and DMSO? If ionomycin treatment does not cause a decrease in WT mitochondria motility, then how are the authors convinced that the reduction in *Cyfp1*^{+/-} mitochondria motility is indeed due to the calcium availability provided by ionomycin? The authors should consider validating the effect of ionomycin in actually causing an increase in intracellular calcium levels.

We thank the reviewer for bringing up this important point. We have validated the effect of ionomycin and BAPTA-AM, and included the data in Figure S5, panel a. With this experiment we could confirm that ionomycin treatment in DIV 3 cortical neurons increases intracellular calcium concentration to a similar extent in both genotypes, while BAPTA-AM effectively reduces it.

Elevated calcium is known to completely block mitochondria movement (Yi et al 2004). However, for this to occur, intracellular calcium should reach levels significantly higher (about a 100 times)

than those observed at resting level (50-100nM) (Yi et al 2004). In our experiments, we aimed to maintain $[Ca^{2+}]_i$ at physiological levels, close to the resting state (as shown by the new experiment in Figure S5 panel a). We believe that due to the relatively low percentage of motile mitochondria in DMSO-treated WT neurons, the moderate increase in intracellular calcium is insufficient to further reduce transport. However, in *Cyfp1*^{+/-} neurons, where motility exceeds physiological levels, the increase in calcium is sufficient to restore mitochondrial transport to WT levels.

Figure S5 panel a: Left, representative images of Fluo-4 AM intensity in the axons of WT and *Cyfp1*^{+/-} DIV 3 cortical neurons after DMSO, Ionomycin or BAPTA-AM treatment. Scale bar 30 μm. Right, axoplasmic calcium concentration $[Ca^{2+}]_i$ (nM) measured using Fluo-4 AM imaging (WT n=5 embryos, *Cyfp1*^{+/-} n=4 embryos; mean ± SEM; Two-Way ANOVA, $F_{(2,21)} = 3.996$, p=0.0338; genotype effect p=0.0008, treatment effect p<0.0001, interaction p=0.0338).

Minor point:

Normalization of qPCR/Western Blot data to the respective WT control should be stated in the methods/figure legends. E.g. for the RT-PCR it is only stated to be normalized to mH3f3, but clearly is also normalized to the WT control.

We have added this information both in the methods section and figure legends.

Reviewer #2 (Remarks to the Author):

This study aims to investigate the impact of *Cyfp1* haploinsufficiency, a gene related to neurodevelopmental disorders (NDD), on neuronal development. The research builds on the authors' previous study published in *Nat Commun* in 2019, which demonstrated that *Cyfp1* haploinsufficiency caused changes in both anatomical and functional connectivity in cortical circuits of mice. The new manuscript delves deeper into the underlying molecular mechanisms, with a focus on axonal development. Importantly, the manuscript suggests that *Cyfp1* plays a crucial role in early axonal development, and insufficient calcium uptake is the critical mechanism that governs this process. *CYFIP1* regulates mRNA stability for specific voltage-gated calcium channel subunits, which explains the reduced calcium concentration in *Cyfp1* mutant cells. However, in order to properly support the manuscript's current conclusions, the data in several places (detailed below) would need substantial strengthening, likely involving new experiments.

Major points:

1. In vivo evidence:

The paper focuses on the evidence that *Cyfp1* haploinsufficiency leads to a reduction in intracellular calcium, which in turn affects mitochondria motility, density, membrane potential, and axonal growth. However, all experiments were conducted *in vitro*. It is important to note that both calcium homeostasis and mitochondrial function are highly sensitive to culture conditions, which raises questions about the relevance of these findings *in vivo*. Therefore, demonstrating with functional experiments the *in vivo* relevance of calcium level and mitochondria features caused by *Cyfp1* haploinsufficiency would greatly strengthen the key conclusions of this paper.

We thank the reviewer for raising this important point. To investigate the relevance *ex vivo* of the mitochondrial defects observed *in vitro*, we have: 1) measured axonal mitochondrial membrane potential *ex vivo* (Figure 3 panel d), and 2) assessed ATP production in WT and *Cyfp1*^{+/-} cortical neurons both *in vitro* and *ex vivo* (Figure S3 panels e, f, and g, and Figure S3 panels h, i and j). To address point 2, please refer to our answer to comment 3 for the *in vitro* part.

To address point 1, we prepared brain slices from P5-6 mice, stained them with TMRE, and measured mitochondrial membrane potential in the corpus callosum of WT and *Cyfp1*^{+/-} mice. Our results reveal a reduction in mitochondrial membrane potential in the axons of *Cyfp1*^{+/-} neurons compared to WT, consistent with our *in vitro* findings (Figure 3, panel d).

Figure 3 panel d: Upper right, illustration depicting the brain region analyzed. Left, representative images showing TMRE intensity in the corpus callosum of organotypic brain slices from WT and *Cyfip1*^{+/-} P5-6 mice. Scale bar 10 μ m. Inset scale bar 5 μ m. Bottom right, histograms showing TMRE intensity expressed as a percentage over WT. (WT n=4 animals, 14 organotypic brain slices, *Cyfip1*^{+/-} n=3, 10 organotypic brain slices. mean \pm SEM; Unpaired t-test, p=0.0015).

2. Cell identity in vitro & cell type-specific pathology:

The authors observed that in *Cyfip1* mutant mice, growth and arborization of axons in the superficial layer neurons were slower compared to control mice (Fig 1 & S1). For later in vitro experiments, the authors used E15.5 embryonic brains, which potentially include both deep layer and superficial layer pyramidal neurons with different birth dates. Therefore, it is unclear which cell types were analyzed in these experiments. Also, it is not clear whether there is any difference in phenotype between deep-layer and superficial-layer neurons due to *Cyfip1* haploinsufficiency.

We thank the reviewer for rising this important point. In our previous publication we reported altered white matter integrity and functional connectivity defects across several axonal tracts and brain regions in *Cyfip1*^{+/-} mice (Dominguez-Iturza et al., 2019). Among those regions, the corpus callosum appears to be the most affected axonal tract and the primary focus of our study in 2019. This tract is composed of axons from callosal projection neurons located in layers 2/3 and layer 5 of the cortex. Additionally, our DTI data (Dominguez-Iturza et al., 2019) reveal significant differences in other tracts and brain regions, such as the cerebral peduncle, the thalamus, and the striatum. These regions contain axons from corticofugal projection neurons located in layers 5 and 6 of the cortex. Based on these findings, we believe that CYFIP1 may regulate the axonal development of neurons from various brain regions, including upper and deep layer projections neuros in the cortex.

To determine the diversity of projection neuron types present in our primary cultures, we performed immunostaining with CTIP2 (marker of deep layer projection neurons) and SATB2 (marker of upper layer projection neurons). We are providing this figure for the reviewers only; however, we would be happy to include it as a supplementary panel if the reviewer thinks it is relevant to the manuscript.

Figure legend: Left, representative images from DIV 3 cortical neurons immunostained for CTIP2 (green), SATB2 (magenta), β III-Tubulin (gray) and DAPI (blue). Scale bar 50 μ m. Right, histogram showing the percentage of CTIP2 and SATB2 positive cells over the total number of nuclei (n = 6 embryos, mean \pm SEM; Mann-Whitney test, p=0.8182).

This experiment confirms a comparable proportion of both projection neuron types. These results suggest that in our *in vitro* experiments, we analyzed both upper and deep layer projection neurons.

While we cannot rule out the possibility that CYFIP1 may exert different roles in diverse projection neurons in the cortex, our current work points to a common role in axonal development. Future studies should investigate potential differences, which would advance our understanding of the role of CYFIP1 in cortical development and its implications in disease.

3. Mitochondria function in Cyfip1 mutant

The authors found reduced mitochondria membrane potential in Cyfip1 mutant through TMRM imaging and concluded that "Our results suggest that CYFIP1 regulates mitochondria functions (lines179-181)". It would be valuable to conduct more in-depth research on mitochondria properties that are related to axonal development (ref 43) in order to strengthen their conclusion. This would also help to understand the difference of this study and a previous one where Cyfip1 haploinsufficient flies displayed higher mitochondria membrane potential and oxygen consumption rate (ref 40).

Following the reviewer's suggestion, we quantified ATP levels in cortical neurons at DIV 3, 7 and 14 as well as in the cortex of WT and *Cyfp1*^{+/-} mice across postnatal development (P5, P15, and P30). Our results show that ATP levels are reduced in the *Cyfp1*^{+/-} primary cortical neurons and cortex only at early stages (P5 and DIV 3) (Figure S3 panels e to g, Figure S3 panels h to j). This data further demonstrates that mitochondrial function is affected in *Cyfp1*^{+/-} cortical neurons both during *in vitro* and *in vivo* development.

As correctly pointed out by the reviewer, previous research from our laboratory has shown that *Cyfp1* haploinsufficient flies display higher mitochondria membrane potential. Those studies were conducted in a different model system, *Drosophila*, on the entire brain and at adult stage and in this case affecting a mitochondrial transporter. We therefore believe that CYFIP1 may regulate mitochondria function/s in different ways through development, ultimately contributing to neuronal dysfunction/s and disease. Future studies should address the precise role of CYFIP1 in regulating mitochondria function across brain regions and developmental stages in different model systems.

Figure S3 panels e, f and g: The histograms represent the quantification of ATP levels in DIV 3 (e), DIV 7 (f), and DIV 14 (g) Wild Type and *Cyfp1*^{+/-} mouse primary cortical neurons, normalized on the protein concentration [$\mu\text{g}/\mu\text{l}$]. (DIV 3, Wild Type n = 9 biological replicates, *Cyfp1*^{+/-} n = 5 biological replicates, mean \pm SEM, Mann-Whitney test p=0.0120; DIV 7, Wild Type n = 9 biological replicates, *Cyfp1*^{+/-} n = 5 biological replicates, mean \pm SEM, Mann-Whitney test p=0.5185; DIV 14, Wild Type n = 9 biological replicates, *Cyfp1*^{+/-} n = 5 biological replicates, mean \pm SEM, Mann-Whitney test p=0.1898).

Fig. S3 panels h, i and j: ATP levels measured by fluorescence intensity. The histograms represent the quantification of ATP levels in P5 (**h**), P15 (**i**), and P30 (**j**) Wild Type and *Cyfip1*^{+/-} mouse cortex, normalized on the tissue's weight. (P5, Wild Type n = 11, *Cyfip1*^{+/-} n = 8, mean ± SEM, Mann-Whitney test p=0.0328; P15, Wild Type n = 5, *Cyfip1*^{+/-} n = 6, mean ± SEM, Mann-Whitney test p=0.4286; P30, Wild Type n = 3, *Cyfip1*^{+/-} n = 5, mean ± SEM, Mann-Whitney test p>0.9999).

4. Developmental changes

The authors conducted experiments on all in vitro parts and identified phenotypes at DIV 3 and 4. However, further investigation is needed to determine whether these phenotypes will continue to persist at a later stage or if they will disappear, similar to the axonal morphology phenotype in vivo. This information could be extremely valuable in advancing our understanding of the pathologies caused by *Cyfip1* haploinsufficiency, as well as the relationships among the variable endophenotypes caused by this condition.

The reviewer raised an excellent point. We have analyzed mitochondrial transport and membrane potential, as well as protein levels of the calcium channels, at DIV 7 and DIV 14 *in vitro*. The new findings have been incorporated into the revised manuscript.

Our results show that mitochondrial transport is unaffected at later developmental stages (Figure S2 panel c). Interestingly, mitochondrial membrane potential seems to increase in *Cyfip1*^{+/-} neurons at DIV 7 but normalizes to WT levels at DIV 14 (Figure S3 panel c and d). Finally, we observed that the protein levels of the calcium channels are similar between WT and *Cyfip1*^{+/-} neurons at DIV 7 and DIV 14 (Figure S4 panel a).

Overall, our results suggest that CYFIP1 regulates mitochondria transport and function during early developmental stages. The interesting observation that mitochondrial membrane potential in *Cyfip1*^{+/-} neurons shifts from downregulated to upregulated during development raises an

important question about the role of CYFIP1 at different developmental time points, which should be further investigated in future studies.

Fig. S2 panel c: Left, representative kymographs showing transport of mitochondria along axons of WT and *Cyfip1*^{+/-} DIV 7 and DIV 14 cortical neurons. Scale bar 50 μ m. Right, percentage of motile mitochondria in WT and *Cyfip1*^{+/-} DIV 7 and DIV 14 cortical neurons (DIV 7: WT n=7 embryos, 33 axons, *Cyfip1*^{+/-} n=6 embryos, 26 axons; mean \pm SEM; Mann-Whitney test, p=0.4452. DIV 14: WT n=9 embryos, 43 axons, *Cyfip1*^{+/-} n=7 embryos, 31 axons; mean \pm SEM; Mann-Whitney test, p=0.4698).

Fig. S3 panel c and d: Left, representative images of TMRE intensity in the cell body of WT and *Cyfip1*^{+/-} DIV 7 (c) and DIV 14 (d) cortical neurons. Scale bar 100 μ m, inset scale bar 50 μ m. Right, Histograms represent average TMRE intensity in WT and *Cyfip1*^{+/-} cell body (DIV 7, WT n=9 embryos, *Cyfip1*^{+/-} n=11 embryos, mean \pm SEM, Mann-Whitney test, p=0.0770; DIV 14, WT n=7 embryos, *Cyfip1*^{+/-} n=7 embryos, mean \pm SEM, Mann-Whitney test, p=0.6439).

Fig. S4 panel a: Left, representative Western Blot showing Ca_v1.2 (CACNA1C), Ca_v2.3 (CACNA1E), Ca_v3.3 (CACNA1I), Ca_vγ2 (CACNG2/Stargazin) and Ca_vβ3 (CACNB3) in membrane-enriched fractions from WT and *Cyfip1*^{+/-} DIV 7 (e) and DIV 14 (f) cortical neurons. The molecular weight of each protein is indicated in kDa. Right, histogram representing Ca_v1.2, Ca_v2.3, Ca_v3.3, Ca_vγ2 and Ca_vβ3 protein expression levels in membrane-enriched fractions from WT and *Cyfip1*^{+/-} DIV 7 (e) and DIV 14 (f) cortical neurons. Protein levels were normalized to Coomassie staining and expressed as a fold change over WT (DIV 7: WT n=3 embryos, *Cyfip1*^{+/-} n=3 embryos; mean ± SEM; Multiple unpaired t-test, Ca_v1.2 p=0.9730, Ca_v2.3 p>0.9999, Ca_v3.3 p=0.6723, Ca_vγ2 p=0.8740, Ca_vβ3 p>0.9999; DIV 14: WT n=3 embryos, *Cyfip1*^{+/-} n=3 embryos; mean ± SEM; Multiple unpaired t-test, Ca_v1.2 p>0.9999, Ca_v2.3 p>0.9999, Ca_v3.3 p=0.9919, Ca_vγ2 p=0.9222, Ca_vβ3 p>0.9999).

Minor points:

5. It would be helpful to annotate which part is axon or dendrites in Figure 2B and 3C.

We have included that information in the y-axis of the histograms or representative images throughout the manuscript.

6. To strengthen the author's argument regarding the focus on axonal mitochondria, it would be beneficial to include additional results that use more rigorous criteria for identifying axons, such as AIS staining, instead of the criteria described by the author in lines 752-754.

Our analysis approach is based on previous studies describing the establishment of neuronal polarity *in vitro*. Neurons in culture exhibit a multipolar morphology until one of the neurites undergoes a period of rapid growth to become the axon, while the others develop into dendrites (Dotti et al., 1988). Based on these findings, it is well established that after few days in culture (2-4 days), the longest neurite of the cell is the axon. To confirm this further, we stained for AnkG, a protein located to the axon initial segment and observed its presence exclusively in the longest

neurite of DIV 3 cortical neurons in culture. We have elaborated on this in the text (lines 666-668) and are providing images of the staining for the reviewer's reference. If the reviewer considers this data is relevant to the manuscript, we would be glad to include it in the supplementary material.

Representative images from DIV 3 cortical neurons immunostained for AnkrinG (yellow), Phalloidin (magenta) and DAPI (blue). Scale bar 20 μ m.

7. It would be helpful to provide evidence that treatments with chemical compounds such as IONO or BAPTA-AM alter cytoplasmic calcium concentration in axons.

We validated the effects of Ionomycin (IONO) and BAPTA-AM on cytoplasmic calcium concentration. We observed that Ionomycin increased axoplasmic calcium concentration, while BAPTA-AM reduced it in both genotypes. This data has been included in Figure S5 new panel a.

Fig. S5 panel a: Left, representative images of Fluo-4 AM intensity in the axons of WT and *Cyfip1*^{+/-} DIV 3 cortical neurons after DMSO, Ionomycin (IONO) or BAPTA-AM treatment. Scale

bar 30 μm . Right, axoplasmic calcium concentration $[\text{Ca}^{2+}]$ (nM) measured using Fluo-4 AM imaging (WT n=5 embryos, *Cyfp1*^{+/-} n=4 embryos; mean \pm SEM; Two-Way ANOVA, $F_{(2,21)} = 3.996$, $p=0.0338$; genotype effect $p=0.0008$, treatment effect $p<0.0001$, interaction $p=0.0338$).

8. Each step of neuronal development and maturation influences the other. It would be valuable to investigate other neuronal developmental parameters, such as dendrite growth, in *Cyfp1* mutants.

We thank the reviewer for this interesting comment. Our lab and other colleagues in the field have previously described that changes in *Cyfp1* levels, whether through silencing or overexpression, affect dendritic and spine development. Specifically, reduced CYFIP1 levels cause decreased dendritic complexity and increased number of immature spines both *in vitro* and *in vivo* (De Rubeis et al., 2013; Pathania et al., 2014).

Reviewer #3 (Remarks to the Author):

In this manuscript, the authors examine the role of Cyfip1, a protein associated with higher risk of ASD and SCZ, in the regulation of axonal growth in cortical neurons using *cyfip1*^{+/-} mice as a model. This is an important question as defects in axonal growth and branching are emerging as common defects in multiple genetic models of ASD, leading to dysfunctional connectivity and behavioral phenotypes. Cyfip1 function has been extensively studied in dendrites and spine formation/plasticity in neurons, but its role in axons is less clear.

Here the authors demonstrated that CPN neurons display a delay in axonal development *in vivo* and *in vitro*, an effect that appears cell autonomous. This defect correlates with an increase in mitochondrial density and motility, as well as a decrease in mitochondrial membrane potential, all signs that are coherent with a delay in axonal maturation. Interestingly, the authors show a decreased concentration in intracellular calcium when CYFIP1 levels are reduced. Mechanistically, they provided evidence of a reduced expression of specific calcium channel subunits due to a decreased stability of their respective mRNAs in *cyfip1*^{+/-} neurons. Finally, they reported that increasing intracellular calcium concentration using ionomycin was sufficient to rescue the defects in axonal growth, mitochondrial motility and mitochondrial membrane potential observed in *cyfip1*^{+/-} neurons.

The work is well written, and the images are clear. The results nicely demonstrate a new role for Cyfip1 in the regulation of intracellular calcium concentration during axonal growth. A few additional experiments should be done to better understand how the phenotypes observed are correlated.

Main points:

1 - It remains unclear to me how an overall increase in mitochondrial density observed in axons from *cyfip1*^{+/-} neurons is not concomitant with an increase in the total mitochondrial mass. Do the authors believe that this effect is due to the same number of mitochondria present in a smaller volume? This conclusion could be confirmed by performing additional experiments and quantify the expression of mitochondrial resident/structural proteins in *cyfip1*^{+/-} neurons compared to control? Maybe by Western blot experiments and normalizing on the cytoskeletal?

We thank the reviewer for this question and apologize if our explanation in the text was unclear. Our results in Figure 2, panels b and c, show that the density of axonal mitochondria is increased in *Cyfip1*^{+/-} neurons compared to WT, while the total mitochondrial mass remains unchanged. We hypothesize that altered axonal transport of mitochondria leads to their accumulation in the axonal compartment without affecting the overall mitochondrial mass of the cell. Supporting this hypothesis, our experiments in Figure 2, panel d, demonstrate increased motility of axonal

mitochondria, which could explain their higher density in axons. We have now included this part in the discussion.

2 - Linked to the previous comment, is mitochondrial morphology affected in *cyfip1*^{+/-} neurons? In some pictures, mitochondria appear smaller in *cyfip1*^{+/-} neurons.

This is a very important point also raised by reviewer #1. We performed electron microscopy analysis of axonal mitochondria in DIV 3 cortical neurons cultured *in vitro* using microfluidic devices and measured mitochondrial length and shape. Please refer to our answer to reviewer #1 comment 2.

3 - Or does it mean that mitochondria are not well distributed in *cyfip1*^{+/-} neurons? If so, would it possible that *cyfip1*^{+/-} neurons display a reduction in mitochondrial density in the somatodendritic compartment? Or maybe mitochondria are not well distributed in axons? The authors should analyze mitochondria density and morphology in axons using microfluidic device, allowing a better resolution of the more distal portion.

The reviewer is correct. Based on our results, we hypothesize that mitochondria have a differential distribution in WT and *Cyfip1*^{+/-} neurons. While the total mitochondrial mass remains constant, *Cyfip1*^{+/-} axons contain higher density of mitochondria. Following the reviewer suggestion we have analyzed mitochondrial density in the dendritic compartment and found no difference between WT and *Cyfip1*^{+/-} neurons. This data has been included in Fig. S2 panel a.

Fig. S2 panel a: Left, representative images of DIV 3 WT and *Cyfip1*^{+/-} cortical neurons after transfection with mito-RFP plasmid (magenta). The number of mitochondria along dendrites was calculated and normalized to the length of each dendrite. The average density of mitochondria per dendrite was determined for each neuron and then averaged by animal. Scale bar 20 μm .

Inset scale bar 10 μm . Right, average mitochondrial density of the entire neuron (WT n=10, and *Cyfp1*^{+/-} n=6; mean \pm SEM; Mann-Whitney test; p=0.8749).

4 - If additional features emerge from this analysis, the authors could test if ionomycin rescues the phenotype in *cyfp1*^{+/-} neurons.

No differences were observed in the distribution of mitochondria in *Cyfp1*^{+/-} dendritic compared to WT.

5 - The defect in mitochondrial motility in *cyfp1*^{+/-} neurons should be better characterized with additional information regarding the directionality (anterograde versus retrograde), the % of oscillatory and % of directed transport.

Following the reviewer suggestion, a point also raised by reviewer 2, we have analyzed mitochondrial length and speed of movement, both in anterograde and retrograde direction. None of these parameters is significantly affected in *Cyfp1*^{+/-} compared with WT. We have included this data in Figure S2 panel b.

6 - The authors suggest that CYFIP1 could control the stability of specific mRNAs through its binding to HuD and/or HuR proteins. The authors should analyze the expression of these two RBPs in *cyfp1*^{+/-} neurons and determine if their expression is changed when CYFIP1 is reduced.

We have performed western blot to analyze the protein expression of HuD and HuR in WT and *Cyfp1*^{+/-} neurons and observed that they remain unchanged (Figure S4 panel g).

To further confirm CYFIP1 regulates the stability of calcium channel mRNAs through HuD and HuR, we have: 1) verified CYFIP1 binding to HuD and HuR by co-immunoprecipitation (Figure S4, panel f), and 2) conducted RNA immunoprecipitation of HuD and HuR in WT and *Cyfp1*^{+/-} cortical neurons to assess their binding of *mCacna1c*, *mCacna1e* and *mCacna1i*. Please refer to our answer to reviewer #1 point 1 for more details.

Fig. S4 panel g: Left, representative Western Blot showing CYFIP1, HuD/Elavl4 and HuR/Elavl1 proteins in WT and *Cyfp1^{+/-}* DIV 3 cortical neurons. The molecular weight of each protein is indicated in kDa. Right, histogram representing HuD/Elavl4, HuR/Elavl1 and CYFIP1 protein expression levels normalized to Coomassie staining and expressed as a fold change over WT (WT n=8, *Cyfp1^{+/-}* n=8; mean \pm SEM; Multiple Mann-Whitney t-test, HuD/Elavl4 p=0.1828, HuR/Elavl1 p=0.2907, CYFIP1 p=0.0116).

Ricci et al.,

RESPONSE TO REFEREES

Reviewer #1 (Remarks to the Author):

Summary

In this manuscript, the authors investigate the significance of cytoplasmic FMR1-interacting protein (Cyfip1) in cortical development. They find a role for Cyfip1 in regulation of mRNA stability of selected transcripts (possibly via its known binding partners HuD and HuR), leading to the loss of subsequent protein expression. They demonstrate that esp. the loss of voltage-sensitive calcium channel subunit expression modulates intracellular calcium levels. This change in calcium availability influences the mitochondrial motility and function, which is suggested to be contribute to cortical axonal development and circuit formation. The authors undertake a straightforward and simple approach for investigating their research questions and report their findings in a coherent manner. However, there is a tendency of the authors drawing strong conclusions from their experiments which do not provide such a wider scope. For the final hypothesis to be more convincing, the authors might need to fill more gaps in their flow of experiments. Overall, the manuscript can benefit from the following suggestions.

Major points:

1. While the authors use a cDNA rescue construct in the beginning, this is not consistently done throughout the manuscript. This is a missed opportunity, as mutation of CYFIP1 that lead to a reduced interaction with the FMRP complex (as in their previous publication PMID 18805096) could serve as the ideal control to shown that their effect of CYFIP1 loss is indeed via interaction with this complex.

We thank the reviewer for this important point. As the reviewer points out, in a previous publication we were able to disentangle the contribution of each CYFIP1 function (actin remodeling or translational control) in regulating spine morphology. However, CYFIP1, in addition to FMRP also interacts with other RNA binding proteins such as HuD, HuR, Caprin etc (De Rubeis et al., 2013) some of those involved in the stability of the mRNAs. In this study, our data suggests that CYFIP1 regulation of intracellular neuronal calcium is independent of those two canonical functions, one of which includes FMRP. Our data indicates that CYFIP1 regulates mRNA stability of calcium channels, through its binding with HuD/Elavl4 and HuR/Elavl1 (Figures 4 and 5), a mechanism that would be independent of FMRP. To further confirm this, we have: 1) verified CYFIP1 binding to HuD and HuR by co-immunoprecipitation (Figure S4, new panel f), and 2) conducted RNA immunoprecipitation of HuD and HuR in WT and *Cyfip1*^{+/-} cortical neurons to assess their binding of *mCacna1c*, *mCacna1e* and *mCacna1i*. Our results demonstrate that both, HuD and HuR, bind *mCacna1c*, *mCacna1e*, and *mCacna1i*. Importantly, we observed reduced binding of HuD and

HuR to *mCacna1c* in *Cyfp1^{+/-}* neurons compared with WT (Figure 5, new panels d and e). This finding supports our hypothesis that HuD/HuR binding to these calcium channel mRNAs may play a role in regulating their stability, which appears to be diminished under *Cyfp1^{+/-}* deficient conditions. Altogether, our data demonstrate that CYFIP1 regulates the stability of calcium channel mRNAs through its interaction with HuD and HuR, rather than controlling their translation via FMRP.

We have included the new panels below for easier reference.

Figure S4 panel f: Representative Western blot of CYFIP1 protein immunoprecipitation from DIV 3 wild-type cortical neurons, showing the detection of HuD/Elavl4 and HuR/Elavl1 proteins.

Figure 5 panels d and e: **d** HuD/Elavl4 RNA immunoprecipitation (RNA-IP) from DIV 3 WT and *Cyfp1^{+/-}* cortical neurons. Histograms represent calcium channel mRNAs enrichment, calculated as the ratio over the positive control (*mDlg4*) (WT n=4, *Cyfp1^{+/-}* n=4; Multiple Mann-Whitney test; *mCacna1c* mRNA p=0.0285, *mCacna1e* mRNA p=0.0285, *mCacna1i* mRNA p=0.8857). **e** HuR/Elavl1 RNA-IP from DIV 3 WT and *Cyfp1^{+/-}* cortical neurons. Histograms represent calcium channel mRNAs enrichment, calculated as the ratio over the positive control (*mDlg4*) (Multiple

Mann-Whitney test; *mCacna1c* mRNA p=0.0310, *mCacna1e* mRNA p=0.2270, *mCacna1i* mRNA p=0.6289).

2. Along similar lines, no experiments directly address whether HuD/HuR are involved in the phenotype of CYFIP^{+/-} neurons, it is only suggested due to the presence of HuD/HuR motifs in the RNAs bound to CYFIP1.

Following the reviewer's suggestion, we have conducted additional experiments demonstrating the following: 1) CYFIP1 binds HuD and HuR (Figure S4, new panel f), and 2) HuD and HuR bind to *mCacna1c*, *mCacna1e* and *mCacna1i* mRNAs, with reduced binding of HuD and HuR to *mCacna1c* and *mCacna1e* in *Cytip1^{+/-}* neurons (Figure 5, panels d and e). These findings further support the notion that CYFIP1 regulates the stability of these calcium channels through its interaction with HuD and HuR.

3. Same for the specific voltage-sensitive calcium channel subunits affected, could overexpression of them also rescue the phenotype?

Following the reviewer's suggestion, to investigate the role of voltage gated calcium channels (VGCCs) in the delayed axonal growth phenotype of *Cytip1^{+/-}* cortical neurons, we treated DIV 3 cortical neurons with the VGCCs agonists Bay-K-8644 and Nefiracetam and monitored their growth over 24h. Bay-K-8644 specifically activates L-type calcium channels, whereas Nefiracetam has a broader affect, targeting both L- and N-type calcium channels. CACNA1C, the most affected channel in *Cytip1^{+/-}* cortical neurons, is an L-type calcium channel, and is therefore activated by both Bay-K-8644 and Nefiracetam. Our results demonstrate that treatment with either agonist effectively rescues the delayed axonal growth phenotype observed in *Cytip1^{+/-}* neurons (Figure 6 new panel d).

We have included the new panel below for easier reference.

Figure 6 panel d: Left, representative images of axons growing in the axonal compartment of microfluidic devices at DIV 3 and DIV 4 for WT and *Cytip1^{+/-}* neurons treated with DMSO, (S)-(-)-Bay-K-8644 (50 μ M) and Nefiracetam (10 μ M). Scale bar 200 μ m. Growth rate of each individual

axon was quantified as represented in the scheme (black line represents DIV 3 and red line DIV 4). Right, average axonal growth rate in WT and *Cyfp1*^{+/-} neurons treated with DMSO, (S)-(-)-Bay-K-8644 and Nefiracetam (WT n=5 embryos, 142 DMSO-treated axons, 190 Bay-K-8644-treated axons and 189 Nefiracetam-treated axons, and *Cyfp1*^{+/-} n=7 embryos, 150 DMSO-treated axons, 258 Bay-K-8644-treated axons and 207 Nefiracetam-treated axons; mean ± SEM; Two-Way ANOVA, $F_{(2,30)} = 5.8$, p=0.0070; genotype effect p=0.0003, treatment effect p<0.0001, interaction p=0.0070).

Additionally, it is important to highlight that VGCCs play a crucial role in generating spontaneous regenerative calcium transients (SRCaTs) in developing cortical neurons, with these transients being particularly pronounced in axons (Kamijo et al 2018). Knockout of endogenous Cav1.2 (*mCacna1c*) in primary cortical neurons has been shown to reduce neurite length, indicating that Cav1.2-mediated calcium signaling promotes neurite elongation (Kamijo et al 2018), and further supporting our hypothesis. This information was already mentioned in the discussion section (lines 449-451).

4. Not only mitochondrial movement, but also mitochondrial shape is sensitive to cellular Ca²⁺ concentration (PMID: 18838687, 29694881). The authors use the number of mitochondria as a measure for their abundance, but the actual mitochondrial mass could be similar if the increased mitochondrial density in CYFIP^{-/-} neurons was made up of smaller mitochondria. The authors should quantify the length of mitochondria and factor this into their conclusions.

We thank the reviewer for rising this crucial point. In response to the reviewer's suggestion, we conducted electron microscopy on DIV 3 cortical neurons *in vitro* and assessed the length and shape of axonal mitochondria. Our findings indicate that both the area and the major axis of mitochondria are increased in *Cyfp1*^{+/-} axons. However, the roundness is not significantly different. Altogether, these results suggest that mitochondria in *Cyfp1*^{+/-} axons are slightly larger and more elongated than in WT. This data has been included in Figure 3 and incorporated into the results and discussion of the manuscript.

Figure 3 panel e: Left, representative electron microscopy images of mitochondria along WT and *Cyfp1^{+/-}* DIV 3 cortical axons. Scale bar 250 nm. Right, histograms showing average mitochondrial area, major axis, minor axis and roundness, expressed as percentage over WT. (WT n=12, *Cyfp1^{+/-}* n=13; mean ± SEM; Mann–Whitney test; mitochondria area p=0.0055, major axis p=0.0398, minor axis p=0.2101, roundness p=0.1519).

5. Next to the reduction in axon length, there appear to be less labelled axon in Fig 1b. Is this a function of the slower growth or is there also some form of neuronal cell death triggered?

The number of labeled axons depends on the number of neurons that were electroporated. During *in utero* electroporation, only a fraction of neuronal progenitors located at the ventricular zone will take up the plasmid. Only neurons derived from those progenitors will contain the plasmid, express tdTomato, and thus be visible for analysis. To ensure our measurements are independent of the number of electroporated neurons or labeled axons, we: in panel a, normalize the number of axons in each bin by the total number of axons in bin 1 (as described in methods). In panel b, axonal growth was measured as a whole, rather than by individual axons, and was normalized to the total length of the cortex.

6. The direction of mitochondrial motility (anterograde vs. retrograde) is thought to be indicative of their health. Considering the lower membrane potential of *Cyfp1^{+/-}* axonal mitochondria, it would be interesting to measure the directionality of mitochondrial movement.

Following the reviewer’s suggestion, we measured both mitochondrial length and speed of movement in anterograde and retrograde directions. None of these parameters showed significant differences between *Cyfp1^{+/-}* and WT. We have included this data in Figure S2 panel b.

Figure S2 panel b: Representative histograms showing forward length, forward speed, backward length, backward speed and reverse direction frequency of mitochondria movement along axons of WT and *Cyfp1^{+/-}* DIV 3 cortical neurons (WT n=11 embryos, 53 axons, *Cyfp1^{+/-}* n=8 embryos,

31 axons; mean \pm SEM; Mann-Whitney test, forward length $p=0.3511$, forward speed $p=0.9678$, backward length $p=0.2723$, backward speed $p=0.6574$, reverse direction frequency $p=0.2375$).

7. Mitochondrial Ca^{2+} uptake also depends on the mitochondrial membrane potential. How does the mitochondrial Ca^{2+} content alter upon loss of CYFIP? Could increased mitochondrial Ca^{2+} uptake underlie the reduction in cytosolic Ca^{2+} and the reduced membrane potential?

To assess mitochondrial Ca^{2+} content, we transfected WT and *Cyfp1*^{+/-} neurons with the pCAG-mitoGCaMP5G plasmid. The analysis of mitochondrial Ca^{2+} levels in axons showed a significant reduction in DIV 3 *Cyfp1*^{+/-} neurons compared to WT (Figure 3 panel b). These findings suggest that diminished mitochondrial Ca^{2+} uptake and membrane potential are most likely a consequence of reduced intracellular calcium levels rather than a primary driving factor.

Figure 3 panel b: Left, representative images of mitoGCaMP5G fluorescence intensity in the axonal mitochondria of WT and *Cyfp1*^{+/-} DIV 3 and DIV 4 cortical neurons. Scale bar 20 μm . Inset scale bar 5 μm . Right, histogram showing the axonal mitochondrial calcium concentration [Ca²⁺] expressed as a percentage over WT (WT $n=8$ embryos, *Cyfp1*^{+/-} $n=8$ embryos; mean \pm SEM; Mann-Whitney test, $p=0.0030$).

8. How do the authors explain the unaltered mitochondrial motility in WT neurons when compared between ionomycin and DMSO? If ionomycin treatment does not cause a decrease in WT mitochondria motility, then how are the authors convinced that the reduction in *Cyfp1*^{+/-} mitochondria motility is indeed due to the calcium availability provided by ionomycin? The authors should consider validating the effect of ionomycin in actually causing an increase in intracellular calcium levels.

We thank the reviewer for bringing up this important point. We have validated the effect of ionomycin and BAPTA-AM, and included the data in Figure S5, panel a. With this experiment we

could confirm that Ionomycin treatment in DIV 3 cortical neurons increases intracellular calcium concentration to a similar extent in both genotypes, while BAPTA-AM effectively reduces it.

Elevated calcium is known to completely block mitochondria movement (Yi et al 2004). However, for this to occur, intracellular calcium should reach levels significantly higher (about a 100 times) than those observed at resting level (50-100nM) (Yi et al 2004). In our experiments, we aimed to maintain $[Ca^{2+}]_i$ at physiological levels, close to the resting state (as shown by the new experiment in Figure S5 panel a). We believe that due to the relatively low percentage of motile mitochondria in DMSO-treated WT neurons, the moderate increase in intracellular calcium is insufficient to further reduce transport. However, in *Cytip1^{+/-}* neurons, where motility exceeds physiological levels, the increase in calcium is sufficient to restore mitochondrial transport to WT levels.

Figure S5 panel a: Left, representative images of Fluo-4 AM intensity in the axons of WT and *Cytip1^{+/-}* DIV 3 cortical neurons after DMSO, Ionomycin or BAPTA-AM treatment. Scale bar 30 μ m. Right, axoplasmic calcium concentration $[Ca^{2+}]_i$ (nM) measured using Fluo-4 AM imaging (WT n=5 embryos, *Cytip1^{+/-}* n=4 embryos; mean \pm SEM; Two-Way ANOVA, $F_{(2,21)} = 3.996$, $p=0.0338$; genotype effect $p=0.0008$, treatment effect $p<0.0001$, interaction $p=0.0338$).

Minor point:

Normalization of qPCR/Western Blot data to the respective WT control should be stated in the methods/figure legends. E.g. for the RT-PCR it is only stated to be normalized to mH3f3, but clearly is also normalized to the WT control.

We have added this information both in the methods section and figure legends.

Reviewer #2 (Remarks to the Author):

This study aims to investigate the impact of *Cyfp1* haploinsufficiency, a gene related to neurodevelopmental disorders (NDD), on neuronal development. The research builds on the authors' previous study published in *Nat Commun* in 2019, which demonstrated that *Cyfp1* haploinsufficiency caused changes in both anatomical and functional connectivity in cortical circuits of mice. The new manuscript delves deeper into the underlying molecular mechanisms, with a focus on axonal development. Importantly, the manuscript suggests that *Cyfp1* plays a crucial role in early axonal development, and insufficient calcium uptake is the critical mechanism that governs this process. *CYFIP1* regulates mRNA stability for specific voltage-gated calcium channel subunits, which explains the reduced calcium concentration in *Cyfp1* mutant cells. However, in order to properly support the manuscript's current conclusions, the data in several places (detailed below) would need substantial strengthening, likely involving new experiments.

Major points:

1. In vivo evidence:

The paper focuses on the evidence that *Cyfp1* haploinsufficiency leads to a reduction in intracellular calcium, which in turn affects mitochondria motility, density, membrane potential, and axonal growth. However, all experiments were conducted *in vitro*. It is important to note that both calcium homeostasis and mitochondrial function are highly sensitive to culture conditions, which raises questions about the relevance of these findings *in vivo*. Therefore, demonstrating with functional experiments the *in vivo* relevance of calcium level and mitochondria features caused by *Cyfp1* haploinsufficiency would greatly strengthen the key conclusions of this paper.

We thank the reviewer for raising this important point. To investigate the relevance *ex vivo* of the mitochondrial defects observed *in vitro*, we have: 1) measured axonal mitochondrial membrane potential *ex vivo* (Figure 3 panel d), and 2) assessed ATP production in WT and *Cyfp1*^{+/-} cortical neurons both *in vitro* and *ex vivo* (Figure S3 panels e, f, and g, and Figure S3 panels h, i and j). To address point 2, please refer to our answer to comment 3 for the *in vitro* part.

To address point 1, we prepared brain slices from P5-6 mice, stained them with TMRE, and measured mitochondrial membrane potential in the corpus callosum of WT and *Cyfp1*^{+/-} mice. Our results reveal a reduction in mitochondrial membrane potential in the axons of *Cyfp1*^{+/-} neurons compared to WT, consistent with our *in vitro* findings (Figure 3, panel d).

Figure 3 panel d: Upper right, illustration depicting the brain region analyzed. Left, representative images showing TMRE intensity in the corpus callosum of organotypic brain slices from WT and *Cyfip1*^{+/-} P5-6 mice. Scale bar 10 μm. Inset scale bar 5 μm. Bottom right, histograms showing TMRE intensity expressed as a percentage over WT. (WT n=4 animals, 14 organotypic brain slices, *Cyfip1*^{+/-} n=3, 10 organotypic brain slices. mean ± SEM; Unpaired t-test, p=0.0015).

2. Cell identity in vitro & cell type-specific pathology:

The authors observed that in *Cyfip1* mutant mice, growth and arborization of axons in the superficial layer neurons were slower compared to control mice (Fig 1 & S1). For later in vitro experiments, the authors used E15.5 embryonic brains, which potentially include both deep layer and superficial layer pyramidal neurons with different birth dates. Therefore, it is unclear which cell types were analyzed in these experiments. Also, it is not clear whether there is any difference in phenotype between deep-layer and superficial-layer neurons due to *Cyfip1* haploinsufficiency.

We thank the reviewer for rising this important point. In our previous publication we reported altered white matter integrity and functional connectivity defects across several axonal tracts and brain regions in *Cyfip1*^{+/-} mice (Dominguez-Iturza et al., 2019). Among those regions, the corpus callosum appears to be the most affected axonal tract and the primary focus of our study in 2019. This tract is composed of axons from callosal projection neurons located in layers 2/3 and layer 5 of the cortex. Additionally, our DTI data (Dominguez-Iturza et al., 2019) reveal significant differences in other tracts and brain regions, such as the cerebral peduncle, the thalamus, and the striatum. These regions contain axons from corticofugal projection neurons located in layers 5 and 6 of the cortex. Based on these findings, we believe that CYFIP1 may regulate the axonal development of neurons from various brain regions, including upper and deep layer projections neurons in the cortex.

To determine the diversity of projection neuron types present in our primary cultures, we performed immunostaining with CTIP2 (marker of deep layer projection neurons) and SATB2 (marker of upper layer projection neurons). We are providing this figure for the reviewers only; however, we would be happy to include it as a supplementary panel if the reviewer thinks it is relevant to the manuscript.

Figure legend: Left, representative images from DIV 3 cortical neurons immunostained for CTIP2 (green), SATB2 (magenta), β III-Tubulin (gray) and DAPI (blue). Scale bar 50 μ m. Right, histogram showing the percentage of CTIP2 and SATB2 positive cells over the total number of nuclei (n = 6 embryos, mean \pm SEM; Mann-Whitney test, p=0.8182).

This experiment confirms a comparable proportion of both projection neuron types. These results suggest that in our *in vitro* experiments, we analyzed both upper and deep layer projection neurons.

While we cannot rule out the possibility that CYFIP1 may exert different roles in diverse projection neurons in the cortex, our current work points to a common role in axonal development. Future studies should investigate potential differences, which would advance our understanding of the role of CYFIP1 in cortical development and its implications in disease.

3. Mitochondria function in Cyfip1 mutant

The authors found reduced mitochondria membrane potential in Cyfip1 mutant through TMRM imaging and concluded that "Our results suggest that CYFIP1 regulates mitochondria functions (lines179-181)". It would be valuable to conduct more in-depth research on mitochondria properties that are related to axonal development (ref 43) in order to strengthen their conclusion. This would also help to understand the difference of this study and a previous one where Cyfip1 haploinsufficient flies displayed higher mitochondria membrane potential and oxygen consumption rate (ref 40).

Following the reviewer's suggestion, we quantified ATP levels in cortical neurons at DIV 3, 7 and 14 as well as in the cortex of WT and *Cyfp1*^{+/-} mice across postnatal development (P5, P15, and P30). Our results show that ATP levels are reduced in the *Cyfp1*^{+/-} primary cortical neurons and cortex only at early stages (P5 and DIV 3) (Figure S3 panels e to g, Figure S3 panels h to j). This data further demonstrates that mitochondrial function is affected in *Cyfp1*^{+/-} cortical neurons both during *in vitro* and *in vivo* development.

As correctly pointed out by the reviewer, previous research from our laboratory has shown that *Cyfp1* haploinsufficient flies display higher mitochondria membrane potential. Those studies were conducted in a different model system, *Drosophila*, on the entire brain and at adult stage and in this case affecting a mitochondrial transporter. We therefore believe that CYFIP1 may regulate mitochondria function/s in different ways through development, ultimately contributing to neuronal dysfunction/s and disease. Future studies should address the precise role of CYFIP1 in regulating mitochondria function across brain regions and developmental stages in different model systems.

Figure S3 panels e, f and g: The histograms represent the quantification of ATP levels in DIV 3 (e), DIV 7 (f), and DIV 14 (g) Wild Type and *Cyfp1*^{+/-} mouse primary cortical neurons, normalized on the protein concentration [$\mu\text{g}/\mu\text{l}$]. (DIV 3, Wild Type n = 9 biological replicates, *Cyfp1*^{+/-} n = 5 biological replicates, mean \pm SEM, Mann-Whitney test p=0.0120; DIV 7, Wild Type n = 9 biological replicates, *Cyfp1*^{+/-} n = 5 biological replicates, mean \pm SEM, Mann-Whitney test p=0.5185; DIV 14, Wild Type n = 9 biological replicates, *Cyfp1*^{+/-} n = 5 biological replicates, mean \pm SEM, Mann-Whitney test p=0.1898).

Fig. S3 panels h, i and j: ATP levels measured by fluorescence intensity. The histograms represent the quantification of ATP levels in P5 (**h**), P15 (**i**), and P30 (**j**) Wild Type and *Cyfip1*^{+/-} mouse cortex, normalized on the tissue's weight. (P5, Wild Type n = 11, *Cyfip1*^{+/-} n = 8, mean ± SEM, Mann-Whitney test p=0.0328; P15, Wild Type n = 5, *Cyfip1*^{+/-} n = 6, mean ± SEM, Mann-Whitney test p=0.4286; P30, Wild Type n = 3, *Cyfip1*^{+/-} n = 5, mean ± SEM, Mann-Whitney test p>0.9999).

4. Developmental changes

The authors conducted experiments on all in vitro parts and identified phenotypes at DIV 3 and 4. However, further investigation is needed to determine whether these phenotypes will continue to persist at a later stage or if they will disappear, similar to the axonal morphology phenotype in vivo. This information could be extremely valuable in advancing our understanding of the pathologies caused by *Cyfip1* haploinsufficiency, as well as the relationships among the variable endophenotypes caused by this condition.

The reviewer raised an excellent point. We have analyzed mitochondrial transport and membrane potential, as well as protein levels of the calcium channels, at DIV 7 and DIV 14 *in vitro*. The new findings have been incorporated into the revised manuscript.

Our results show that mitochondrial transport is unaffected at later developmental stages (Figure S2 panel c). Interestingly, mitochondrial membrane potential seems to increase in *Cyfip1*^{+/-} neurons at DIV 7 but normalizes to WT levels at DIV 14 (Figure S3 panel c and d). Finally, we observed that the protein levels of the calcium channels are similar between WT and *Cyfip1*^{+/-} neurons at DIV 7 and DIV 14 (Figure S4 panel a).

Overall, our results suggest that CYFIP1 regulates mitochondria transport and function during early developmental stages. The interesting observation that mitochondrial membrane potential in *Cyfip1*^{+/-} neurons shifts from downregulated to upregulated during development raises an

important question about the role of CYFIP1 at different developmental time points, which should be further investigated in future studies.

Fig. S2 panel c: Left, representative kymographs showing transport of mitochondria along axons of WT and *Cyfip1*^{+/-} DIV 7 and DIV 14 cortical neurons. Scale bar 50 μ m. Right, percentage of motile mitochondria in WT and *Cyfip1*^{+/-} DIV 7 and DIV 14 cortical neurons (DIV 7: WT n=7 embryos, 33 axons, *Cyfip1*^{+/-} n=6 embryos, 26 axons; mean \pm SEM; Mann-Whitney test, p=0.4452. DIV 14: WT n=9 embryos, 43 axons, *Cyfip1*^{+/-} n=7 embryos, 31 axons; mean \pm SEM; Mann-Whitney test, p=0.4698).

Fig. S3 panel c and d: Left, representative images of TMRE intensity in the cell body of WT and *Cyfip1*^{+/-} DIV 7 (c) and DIV 14 (d) cortical neurons. Scale bar 100 μ m, inset scale bar 50 μ m. Right, Histograms represent average TMRE intensity in WT and *Cyfip1*^{+/-} cell body (DIV 7, WT n=9 embryos, *Cyfip1*^{+/-} n=11 embryos, mean \pm SEM, Mann-Whitney test, p=0.0770; DIV 14, WT n=7 embryos, *Cyfip1*^{+/-} n=7 embryos, mean \pm SEM, Mann-Whitney test, p=0.6439).

Fig. S4 panel a: Left, representative Western Blot showing Ca_v1.2 (CACNA1C), Ca_v2.3 (CACNA1E), Ca_v3.3 (CACNA1I), Ca_vγ2 (CACNG2/Stargazin) and Ca_vβ3 (CACNB3) in membrane-enriched fractions from WT and *Cyfip1*^{+/-} DIV 7 (e) and DIV 14 (f) cortical neurons. The molecular weight of each protein is indicated in kDa. Right, histogram representing Ca_v1.2, Ca_v2.3, Ca_v3.3, Ca_vγ2 and Ca_vβ3 protein expression levels in membrane-enriched fractions from WT and *Cyfip1*^{+/-} DIV 7 (e) and DIV 14 (f) cortical neurons. Protein levels were normalized to Coomassie staining and expressed as a fold change over WT (DIV 7: WT n=3 embryos, *Cyfip1*^{+/-} n=3 embryos; mean ± SEM; Multiple unpaired t-test, Ca_v1.2 p=0.9730, Ca_v2.3 p>0.9999, Ca_v3.3 p=0.6723, Ca_vγ2 p=0.8740, Ca_vβ3 p>0.9999; DIV 14: WT n=3 embryos, *Cyfip1*^{+/-} n=3 embryos; mean ± SEM; Multiple unpaired t-test, Ca_v1.2 p>0.9999, Ca_v2.3 p>0.9999, Ca_v3.3 p=0.9919, Ca_vγ2 p=0.9222, Ca_vβ3 p>0.9999).

Minor points:

5. It would be helpful to annotate which part is axon or dendrites in Figure 2B and 3C.

We have included that information in the y-axis of the histograms or representative images throughout the manuscript.

6. To strengthen the author's argument regarding the focus on axonal mitochondria, it would be beneficial to include additional results that use more rigorous criteria for identifying axons, such as AIS staining, instead of the criteria described by the author in lines 752-754.

Our analysis approach is based on previous studies describing the establishment of neuronal polarity *in vitro*. Neurons in culture exhibit a multipolar morphology until one of the neurites undergoes a period of rapid growth to become the axon, while the others develop into dendrites (Dotti et al., 1988). Based on these findings, it is well established that after few days in culture (2-4 days), the longest neurite of the cell is the axon. To confirm this further, we stained for AnkG, a protein located to the axon initial segment and observed its presence exclusively in the longest

neurite of DIV 3 cortical neurons in culture. We have elaborated on this in the text (lines 666-668) and are providing images of the staining for the reviewer's reference. If the reviewer considers this data is relevant to the manuscript, we would be glad to include it in the supplementary material.

Representative images from DIV 3 cortical neurons immunostained for AnkrinG (yellow), Phalloidin (magenta) and DAPI (blue). Scale bar 20 μ m.

7. It would be helpful to provide evidence that treatments with chemical compounds such as IONO or BAPTA-AM alter cytoplasmic calcium concentration in axons.

We validated the effects of Ionomycin (IONO) and BAPTA-AM on cytoplasmic calcium concentration. We observed that Ionomycin increased axoplasmic calcium concentration, while BAPTA-AM reduced it in both genotypes. This data has been included in Figure S5 new panel a.

Fig. S5 panel a: Left, representative images of Fluo-4 AM intensity in the axons of WT and *Cyfip1^{+/-}* DIV 3 cortical neurons after DMSO, Ionomycin (IONO) or BAPTA-AM treatment. Scale

bar 30 μm . Right, axoplasmic calcium concentration $[\text{Ca}^{2+}]$ (nM) measured using Fluo-4 AM imaging (WT n=5 embryos, *Cyfp1*^{+/-} n=4 embryos; mean \pm SEM; Two-Way ANOVA, $F_{(2,21)} = 3.996$, $p=0.0338$; genotype effect $p=0.0008$, treatment effect $p<0.0001$, interaction $p=0.0338$).

8. Each step of neuronal development and maturation influences the other. It would be valuable to investigate other neuronal developmental parameters, such as dendrite growth, in *Cyfp1* mutants.

We thank the reviewer for this interesting comment. Our lab and other colleagues in the field have previously described that changes in *Cyfp1* levels, whether through silencing or overexpression, affect dendritic and spine development. Specifically, reduced CYFIP1 levels cause decreased dendritic complexity and increased number of immature spines both *in vitro* and *in vivo* (De Rubeis et al., 2013; Pathania et al., 2014).

Reviewer #3 (Remarks to the Author):

In this manuscript, the authors examine the role of Cyfip1, a protein associated with higher risk of ASD and SCZ, in the regulation of axonal growth in cortical neurons using *cyfip1*^{+/-} mice as a model. This is an important question as defects in axonal growth and branching are emerging as common defects in multiple genetic models of ASD, leading to dysfunctional connectivity and behavioral phenotypes. Cyfip1 function has been extensively studied in dendrites and spine formation/plasticity in neurons, but its role in axons is less clear.

Here the authors demonstrated that CPN neurons display a delay in axonal development *in vivo* and *in vitro*, an effect that appears cell autonomous. This defect correlates with an increase in mitochondrial density and motility, as well as a decrease in mitochondrial membrane potential, all signs that are coherent with a delay in axonal maturation. Interestingly, the authors show a decreased concentration in intracellular calcium when CYFIP1 levels are reduced. Mechanistically, they provided evidence of a reduced expression of specific calcium channel subunits due to a decreased stability of their respective mRNAs in *cyfip1*^{+/-} neurons. Finally, they reported that increasing intracellular calcium concentration using ionomycin was sufficient to rescue the defects in axonal growth, mitochondrial motility and mitochondrial membrane potential observed in *cyfip1*^{+/-} neurons.

The work is well written, and the images are clear. The results nicely demonstrate a new role for Cyfip1 in the regulation of intracellular calcium concentration during axonal growth. A few additional experiments should be done to better understand how the phenotypes observed are correlated.

Main points:

1 - It remains unclear to me how an overall increase in mitochondrial density observed in axons from *cyfip1*^{+/-} neurons is not concomitant with an increase in the total mitochondrial mass. Do the authors believe that this effect is due to the same number of mitochondria present in a smaller volume? This conclusion could be confirmed by performing additional experiments and quantify the expression of mitochondrial resident/structural proteins in *cyfip1*^{+/-} neurons compared to control? Maybe by Western blot experiments and normalizing on the cytoskeletal?

We thank the reviewer for this question and apologize if our explanation in the text was unclear. Our results in Figure 2, panels b and c, show that the density of axonal mitochondria is increased in *Cyfip1*^{+/-} neurons compared to WT, while the total mitochondrial mass remains unchanged. We hypothesize that altered axonal transport of mitochondria leads to their accumulation in the axonal compartment without affecting the overall mitochondrial mass of the cell. Supporting this hypothesis, our experiments in Figure 2, panel d, demonstrate increased motility of axonal

mitochondria, which could explain their higher density in axons. We have now included this part in the discussion.

2 - Linked to the previous comment, is mitochondrial morphology affected in *cyfip1*^{+/-} neurons? In some pictures, mitochondria appear smaller in *cyfip1*^{+/-} neurons.

This is a very important point also raised by reviewer #1. We performed electron microscopy analysis of axonal mitochondria in DIV 3 cortical neurons cultured *in vitro* using microfluidic devices and measured mitochondrial length and shape. Please refer to our answer to reviewer #1 comment 2.

3 - Or does it mean that mitochondria are not well distributed in *cyfip1*^{+/-} neurons? If so, would it possible that *cyfip1*^{+/-} neurons display a reduction in mitochondrial density in the somatodendritic compartment? Or maybe mitochondria are not well distributed in axons? The authors should analyze mitochondria density and morphology in axons using microfluidic device, allowing a better resolution of the more distal portion.

The reviewer is correct. Based on our results, we hypothesize that mitochondria have a differential distribution in WT and *Cyfip1*^{+/-} neurons. While the total mitochondrial mass remains constant, *Cyfip1*^{+/-} axons contain higher density of mitochondria. Following the reviewer suggestion we have analyzed mitochondrial density in the dendritic compartment and found no difference between WT and *Cyfip1*^{+/-} neurons. This data has been included in Fig. S2 panel a.

Fig. S2 panel a: Left, representative images of DIV 3 WT and *Cyfip1*^{+/-} cortical neurons after transfection with mito-RFP plasmid (magenta). The number of mitochondria along dendrites was calculated and normalized to the length of each dendrite. The average density of mitochondria per dendrite was determined for each neuron and then averaged by animal. Scale bar 20 μm.

Inset scale bar 10 μm . Right, average mitochondrial density of the entire neuron (WT n=10, and *Cyfp1*^{+/-} n=6; mean \pm SEM; Mann-Whitney test; p=0.8749).

4 - If additional features emerge from this analysis, the authors could test if ionomycin rescues the phenotype in *cyfp1*^{+/-} neurons.

No differences were observed in the distribution of mitochondria in *Cyfp1*^{+/-} dendritic compared to WT.

5 - The defect in mitochondrial motility in *cyfp1*^{+/-} neurons should be better characterized with additional information regarding the directionality (anterograde versus retrograde), the % of oscillatory and % of directed transport.

Following the reviewer suggestion, a point also raised by reviewer 2, we have analyzed mitochondrial length and speed of movement, both in anterograde and retrograde direction. None of these parameters is significantly affected in *Cyfp1*^{+/-} compared with WT. We have included this data in Figure S2 panel b.

6 - The authors suggest that CYFIP1 could control the stability of specific mRNAs through its binding to HuD and/or HuR proteins. The authors should analyze the expression of these two RBPs in *cyfp1*^{+/-} neurons and determine if their expression is changed when CYFIP1 is reduced.

We have performed western blot to analyze the protein expression of HuD and HuR in WT and *Cyfp1*^{+/-} neurons and observed that they remain unchanged (Figure S4 panel g).

To further confirm CYFIP1 regulates the stability of calcium channel mRNAs through HuD and HuR, we have: 1) verified CYFIP1 binding to HuD and HuR by co-immunoprecipitation (Figure S4, panel f), and 2) conducted RNA immunoprecipitation of HuD and HuR in WT and *Cyfp1*^{+/-} cortical neurons to assess their binding of *mCacna1c*, *mCacna1e* and *mCacna1i*. Please refer to our answer to reviewer #1 point 1 for more details.

Fig. S4 panel g: Left, representative Western Blot showing CYFIP1, HuD/Elavl4 and HuR/Elavl1 proteins in WT and *Cyfp1^{+/-}* DIV 3 cortical neurons. The molecular weight of each protein is indicated in kDa. Right, histogram representing HuD/Elavl4, HuR/Elavl1 and CYFIP1 protein expression levels normalized to Coomassie staining and expressed as a fold change over WT (WT n=8, *Cyfp1^{+/-}* n=8; mean \pm SEM; Multiple Mann-Whitney t-test, HuD/Elavl4 p=0.1828, HuR/Elavl1 p=0.2907, CYFIP1 p=0.0116).

Ricci et al.,

RESPONSE TO REFEREES – SECOND REVISION

Reviewer #1 (Remarks to the Author):

The authors have responded to all comments adequately. This is a beautiful study and I recommend accepting the manuscript.

Reviewer #2 (Remarks to the Author):

The authors have addressed all of our questions, and the paper has been significantly improved. Therefore, it is now suitable for publication.

Reviewer #3 (Remarks to the Author):

I have carefully read the authors' responses and the revised manuscript. I am satisfied that the authors have adequately addressed all of my main concerns. The revisions have improved the clarity and quality of the work.

I therefore recommend that the manuscript be accepted for publication.

Reviewer #4 (Remarks to the Author):

We sincerely thank all the reviewers for their time, effort, and valuable feedback. Their insightful comments and constructive suggestions have greatly strengthened and improved our study.